# Techno-economic modelling of hybrid energy system to overcome the load shedding problem: A case study of Pakistan

**Muhammad Paend Bakht** [1,2]*, **Zainal Salam**[1], **Abdul Rauf Bhatti** [3], **Usman Ullah Sheikh**[1], **Nuzhat Khan**[4], **Waqas Anjum** [1,5]

**1** Centre of Electrical Energy Systems, School of Electrical Engineering, Universiti Teknologi Malaysia (UTM), Johor Baharu, Malaysia, **2** Department of Electrical Engineering, Balochistan University of Information Technology, Engineering and Management Sciences (BUITMS), Quetta, Pakistan, **3** Department of Electrical Engineering, Government College University Faisalabad (GCUF), Faisalabad, Pakistan, **4** University Sains Malaysia (USM), Penang, Malaysia, **5** The Islamia University of Bahawalpur, Bahawalpur, Pakistan

* muhammad.paend@buitms.edu.pk

## Abstract

This paper demonstrates the application of hybrid energy system (HES) that comprises of photovoltaic (PV) array, battery storage system (BSS) and stand-by diesel generator (DGen) to mitigate the problem of load shedding. The main work involves techno-economic modelling to optimize the size of HES such that the levelized cost of electricity (LCOE) is minimized. The particle swarm optimization (PSO) algorithm is used to determine the optimum size of the components (PV, BSS). Simulations are performed in MATLAB using real dataset of irradiance, temperature and load shedding schedule of the small residential community situated in the city of Quetta, Pakistan. The LCOE for the HES system under study is 8.32 cents/kWh—which is lower than the conventional load shedding solution, namely the uninterruptable power supply (UPS) (13.06 cents/kWh) and diesel and generator system (29.19 cents/kWh). In fact, the LCOE of the HRES is lower than the grid electricity price of Pakistan (9.3 cents/kWh). Besides that, the HES alleviates the grid burden by 47.9% and 13.1% compared to the solution using the UPS and generator, respectively. The outcomes of the study suggests that HES is able to improve reliability and availability of electric power for regions that is affected by the load shedding issue.

## 1. Introduction

Energy deficiency is a prominent problem for many developing countries, which is caused due to various reasons such as shortage of generation, inefficient power transmission and inadequate or outdated distribution equipment [1]. As a result, the utility operators in the affected countries implement load shedding—a regimented operating condition whereby the grid supply is disconnected from customers within a specified region for several hours per day. In essence, it removes or curtails a certain amount of load when the demand for electricity exceeds the supply capability of the network [2]. The idea is to minimize the deficit between

energy (power division) Pakistan. Data can be accessed from following URLs https://solcast.com/historical-and-tmy/ (Meteorological data) http://ccms.pitc.com.pk/FeederDetails (Local feeder load shedding schedule).

**Funding:** The author(s) received no specific funding for this work.

**Competing interests:** NO authors have competing interests.

generation capacity and demand while ensuring a fair level of supply availability for all consumers [3]. Load shedding is necessary to prevent systemic power failure from system instabilities and the possible costly systemic failures [4]. However, the impact of the load shedding on the long-term basis can be catastrophic for the countries as it severely affects their economic growth. According to the World Bank, regular electricity shortages have badly hurt the economy of several nations, including Pakistan, Sri Lanka, South Africa and India [5].

For Pakistan, although the critical energy situation was recognized in 2007, the roots of the crippling problem can be traced back to the 1990's. The main sources of electric power (total annual generation of 23 GW) were hydroelectric (54%) and the thermal plants (46%) [6]. Over time, the gap between generation and demand widened; by 2018, the shortfall was reported around 3 to 5 GW [7]. The supply-demand unbalances further deteriorated due to the high-energy consumption during the summer, with regard to extensive use of air conditioning. In the winter, the low water flow in major rivers creates problems for the hydro plants. Furthermore, the volatility of the oil prices also caused electricity generation using fossil to be more expensive.

Due to the government's commitment, the generation capacity of the conventional power sources has significantly increased over the past few years. Gradually, the dilemma between power supply and demand eased, but during the summer seasons, a large section of the country still experience power shortage (due to energy rationing). As mentioned previously, the reason for the shortage was insufficient capacity; this is termed as the "hard" power shortage [8]. However, since 2020, the energy crisis in Pakistan has shifted towards the "soft" shortage. In this case, the unavailability is not due to the insufficient installed capacity; rather, it is a result of under-utilization of the generating units due to the lack of supporting facilities, low system efficiency and improper coordination [9, 10]. Although the generation has been increased substantially, little attention is given to upgrade entire chain of the electricity infrastructure, namely the transmission and distribution. Besides, there is a substantial need to relook into the costs recovery mechanism to minimize high circular debt [11]. The circular debt is the shortfall of payments between power sector institutions [12]. The vicious cycle of the debt that emerged in 2006 has been rising continuously. Recently, it has reached to 116.3 billion USD in 2021 (News).

In light of these facts, the electricity shortage remains problematic at the distribution level. As a result, multiple hours of scheduled load shedding is still a common practice by utility operators. Since the permanent solution is not expected in near future, there is a need to resolve this problem for the end-users, at least in the short or medium term. Against this background, the HES is being proposed as an innovative concept to deal with the load shedding problem. A HES is made up of two or more power plants of different types supplied with corresponding fuels (fossil or renewable). Unlike single-source systems, HESs are more reliable, more efficient, and generate power at lower costs [13]. Besides the localised implementation of HES will provide a wide variety of economic and societal benefits. Some indirect benefits include work opportunities, sustainable industrialization, consumer choice, technological advancements and improved healthcare facilities [14].

Considering the Pakistani context, numerous analysis on the various sources combination of HES, optimization techniques and planning criteria are reported in the literature [15–21]. However, there are certain limitations of existing studies, which are discussed below. In [15–18], the focus is for the electrification of remote rural communities as a standalone system. Due to the inherent off-grid architecture, the analysis presented do not consider the load shedding conditions from utility grid. On the other side, the grid connected analysis of HES are performed in [19–21]. The authors in [19] presented the techno-economic analysis of solar-wind-biomass hybrid system, but emphasized toward the economic performance of the

designed system. Their results showed that cost of energy for grid connected hybrid system is lower compared to off-grid system with similar load profiles. Similarly in [20], the feasibility study of grid-tied, PV battery system was analyzed for a household. The study concluded that PV battery system tied with grid is economically and environmentally more feasible compared to unreliable grid with battery system and standalone PV battery system. In both studies however, the technical performance and reliability of their system were not given the due attention. Besides, the control operation of a HES with a PV battery and diesel generator backup system is described in [21], recently. The study presented a good analysis of the instantaneous response of energy sources during recurring scenario of grid outage. Contrary to [19, 20], this work investigated the technical performance of the proposed system while the economic analysis and the cost of energy from energy sources are not analysed.

Despite the above-mentioned studies, the techno-economic aspects of the HES during load shedding is rarely analysed. Specifically, no effort is made to evaluate and compare the performance of the HES with the popular solutions presently adopted by consumers, namely the diesel generator and uninterruptable power supply (UPS) battery-backed system. Literature search reveals a lack of data on the economic benefits of installing HES under Pakistan net metering policy, feed-in tariff (FiT) incentive, and time-of-use (TOU) electricity rates, making it difficult for decision-makers to foster appropriate conditions to attract private investments. Thus, there is a need to present the techno-economic modelling of HES in conjunction with these scenarios. Such an analysis could provide a rationale for electricity supply security (availability) enhancing investments and energy policy decisions.

The main objective of this study is to ascertain the effectiveness of renewables in form of proposed HES to deal with the present load shedding problem. The techno-economic modelling includes optimal system configuration, sizing and the costs of the components while analysing the reliability of the designed system under different weather and load shedding conditions. To examine the real benefits of HES, the actual values of irradiance, temperature, and load shedding schedule of the selected location in Pakistan are taken into account. Outcomes of this study can help improve project designs and increase the certainty for investment decisions.

The organization of the rest of the article is given as: Section 2 introduces the materials and methods applied to accomplish the proposed research objectives. Section 3 illustrates the results along with discussion of the obtained results. Finally, section 4 concludes the paper and provides the recommendations for policy makers.

## 2. Materials and methods

The methodology adopted to carry out the proposed research is given in Fig 1. The modelling process commences with the collection of the weather and load shedding data (data mining) from the specified location. The technical and economic specifications of the HES are also defined. In Step 2, the HES is modelled with help of various sub-models, namely the PV, BSS, inverter and diesel generator. In addition, the utility grid is incorporated into the model to represent the behaviour of the grid during load shedding. Modelling process integrates both technical and economic models of HES components. Besides, the energy management scheme is designed to ensure optimum energy flow under different operating scenarios. In step 3, PSO algorithm is applied to determine the optimal size of HES components ($N_{PV}$, $N_{Bat}$) under specific design constraints. In the last step, comparative evaluation with two conventional solutions is carried out. These are diesel generators and UPS. Finally, the impact of FiT on HES is analysed. All simulations are performed on Windows 10 Pro 64-bit Intel (R) Core (TM) i7-2600 CPU @ 3.40GHz, 3.40 GHz with 8 GB of RAM.

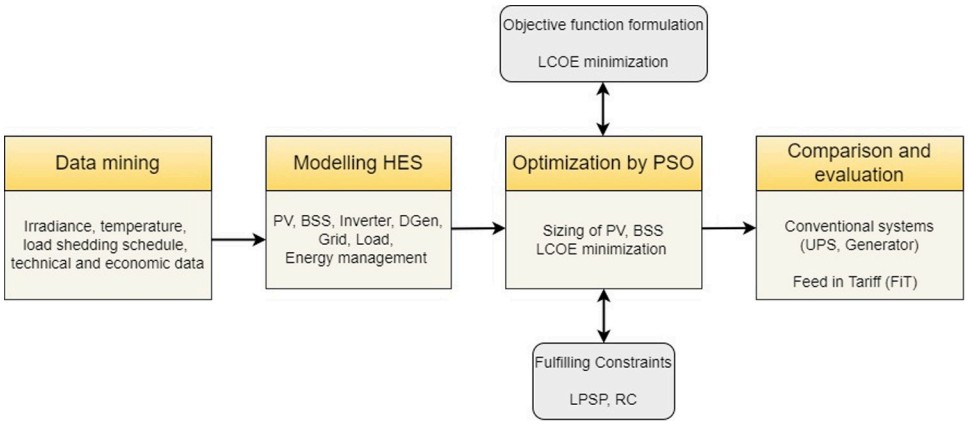

**Fig 1. Block diagram of research methodology.**

## 2.1 HES configuration

The energy system model used in this work is presented in Fig 2. The model integrates the renewable source (PV), storage (BSS) and back-up generator (DGen) to the existing grid at the distribution feeder level. The PV and BSS are connected to the utility grid via a dc bus. The

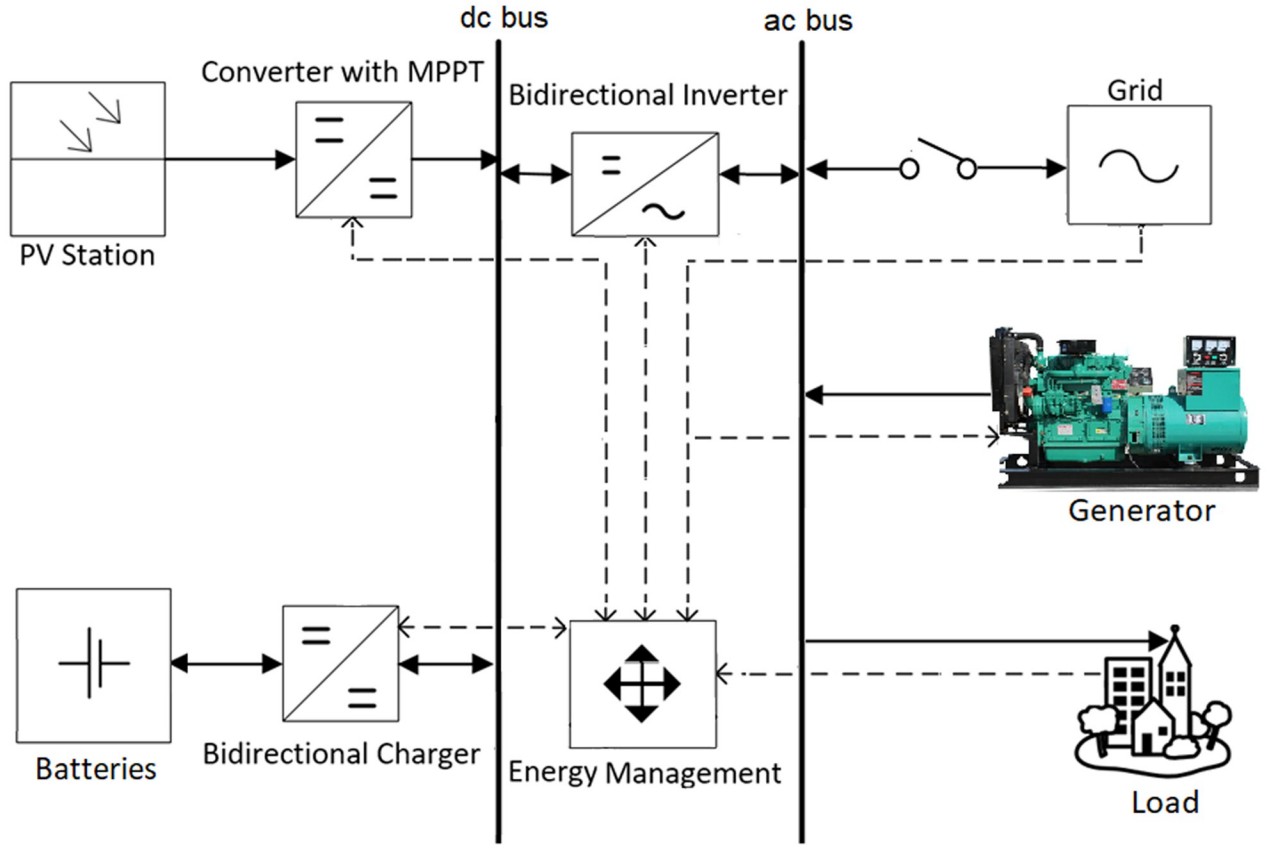

**Fig 2. Architecture of HES.**

dc-dc converter performs the maximum power point tracking (MPPT) control, while the bi-directional charger of BSS manages the flow of dc power in both directions. Lithium-ion batteries are utilized because of their high efficiency, high energy density, extended life time (high number of deep discharge cycle) and faster charging rate [22]. The utility grid, community load (ac), and DGen are directly connected to the ac bus. The ac and dc sources are interfaced through a single bidirectional inverter [23]. A central controller is embedded with EMS to maintain the power balance while satisfying load demand under a number of operational constraints. In the conventional solution for load shedding, the PV and BSS are not available; the effect of the shedding is mitigated mainly by the DGen, and to lesser extent the UPS (with its own local battery).

## 2.2 The techno-economic modelling approach

The concept of techno-economic modelling integrates both the technical and economic aspects of a system in a single mathematical formulation and solution. The technical model computes the power generated and consumed by the system, while the economic model estimates the effective cost of the generated power. These two models are linked by a common denominator—in this case, the LCOE. The techno-economic approach considers the effect of variables from both aspects co-currently, thus, making the solution to be inherently more comprehensive and inclusive. This model will be used to evaluate the overall viability of the HES. It is also used to compare the performance of HES with the UPS and DGen system, presently being used to overcome shedding.

## 2.3 The technical modelling

The components to be modelled are the PV, BSS, inverter, diesel generator and the utility grid. The equation for the power balance of HES is given as

$$((P_{PV}(t) \pm P_{BSS}(t)) \times \eta_{inv}) + P_{DGen}(t) \pm P_{Grid}(t) = P_{Load}(t) \tag{1}$$

Where $P_{Load}(t)$ is the hourly load demand of the system and $P_{PV}(t)$, $P_{BSS}(t)$, $P_{Gen}(t)$ and $P_{Grid}(t)$, are the hourly powers supplied by PV, BSS, diesel generator and the grid, respectively. The negative sign with BSS and grid indicates the capability of power being absorbed by these sources. The $\eta_{inv}$ is the efficiency of the bidirectional inverter.

**2.3.1 PV technical model.** The PV power is calculated using the single diode model [24]. The inputs for the model are irradiance (in W/m$^2$), temperature (in ˚C). The model itself requires the computation of two parameters, namely the series ($R_S$) and shunt ($R_P$) resistances, respectively. The output current of PV cell can be written as

$$I_{PV} = [(I_{SC-STC} + k_i(T - T_{STC}))\frac{G}{G_{STC}}] - I_0[(e)^{\left(\frac{V_{PV} + I_{PV}R_S}{V_T}\right)} - 1] - (\frac{V_{PV} + I_{PV}R_S}{R_P}) \tag{2}$$

The first part of Eq (2), i.e. [($I_{SC\_STC}$ + $k_i$ (T-$T_{STC}$)) G/$G_{STC}$] is the photocurrent that is generated due to the incidence of light. The second part: $I_0$ [e $^{(Vpv+IpvRs) / V}T$ -1], represents the diode current, while the last part (($V_{PV}$+$I_{PV}Rs$) /$R_P$) is the leakage current in shunt resistor. The $I_{SC\text{-}STC}$ is the short circuit current (in amperes) at the standard test condition (STC), while $k_i$ is the short circuit current coefficient (given by the manufacturer), G is the cell surface irradiance whereas $G_{STC}$ is the irradiance at STC (1,000 W/m$^2$), T is the temperature of module (in Kelvin), $T_{STC}$ is the temperature at STC (298$^o$K or 25˚C). The thermal voltage ($V_T$) is equal to $akT/q$ where q represents electron charge and a refers to the ideality factor (1≤ a ≤ 2) and k is the Boltzmann constant. Meanwhile, $R_S$ and $R_P$ represent series and shunt resistances of cell,

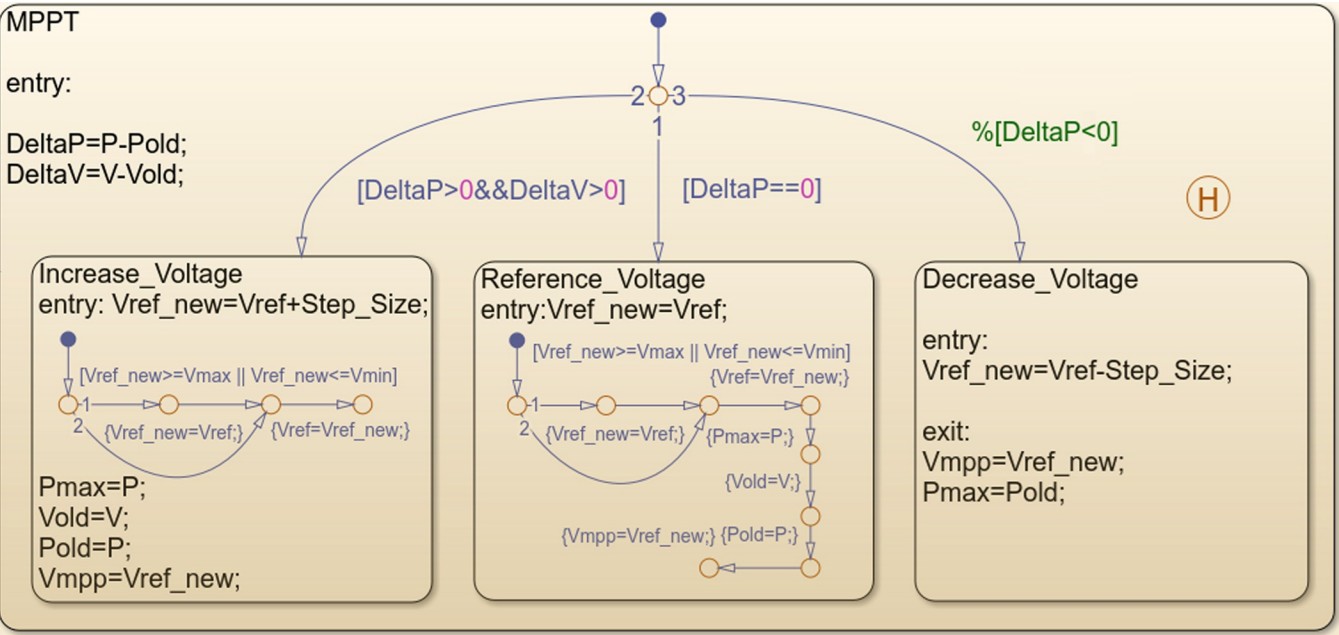

**Fig 3. Stateflow chart of the P&O MPPT algorithm.**

respectively. Due to the transcendental nature of (2), the Newton-Raphson method is employed by varying $V_{PV}$ from zero to the open circuit voltage ($V_{OC}$).

Further, the *P-V* curve is continuously tracked to extract the maximum power, i.e., $P_{max} = V_{mpp} \times I_{PV}$. The perturb and observe (P&O) maximum power point tracker (MPPT) is used to compute $P_{max}$ for a given value of *G* and *T*. In this work, the P&O is implemented using the Stateflow software from Simulink [25]. The Stateflow chart for the P&O chart is shown in Fig 3. Considering the array is uniformly irradiated, the output power of entire system ($P_{PV}$) is calculated by multiplying the total number of modules ($N_{PV}$) with $P_{max}$. The power is modelled on hourly basis according to the actual values of *G* and *T* [26]. Other PV parameters considered are given in [27].

**2.3.2 BSS technical model.** The present value of state of charge (SOC) determines the allowable charging and discharging of the BSS. The SOC of the BSS at any time (*t*) is given by [28] as

$$SOC(t) = SOC(t-1) \times (1 - \delta_{Bat}(t)) + \left(\frac{P_{BSS}(t)}{V_{bus}}\right) \times \eta_{Bat} \times \Delta t \qquad (3)$$

Where $\Delta t$ is the time step for the iteration (one hour), while $\delta_{Bat}$ represents hourly self-discharge factor. For simplicity, $\delta_{Bat}$ is assumed to be zero (ideal battery). The $\eta_{Bat}$ is the battery efficiency (considered 100%). The voltage of dc bus ($V_{bus}$) is taken as 480 V [29]. The lower and upper limits of SOC are labeled as $SOC_L$ and $SOC_U$, respectively. The BSS discharging is halted if $SOC_L$ is reached. The $P_{BSS}(t)$ represents the charging and discharging power of BSS according to required and available power, respectively. The required power to charge the BSS for one time-step (hour) is represented by plus sign and is written as

$$+P_{BSS}(t) = \frac{(SOC_U - SOC(t)) \times C_{Bat} \times N_{Bat}}{\Delta t} \qquad (4)$$

Accordingly, the negative signed $P_{BSS}(t)$ is the maximum power that BSS can continuously deliver (discharge) before reaching the $SOC_L$ limit, i.e.

$$-P_{BSS}(t) = \frac{(SOC(t) - SOC_L) \times C_{Bat} \times N_{Bat}}{\Delta t} \tag{5}$$

Here, $C_{bat}$ represents the rated capacity (of a single unit battery) in kWh. Thus, $N_{bat}$ indicates the total number of batteries in the BSS. To ensure batteries durability, the $SOC$ should be confined within $SOC_U$ and $SOC_L$, which are set to 90% and 10%, respectively [30].

**2.3.3. Inverter technical model.** The inverter is modelled as proposed in [31]. If $P_{Load_{Peak}}$ is the peak load demand of the system and $\eta_{inv}$ is the efficiency of the inverter, the output power of the inverter (in kW) is given by

$$P_{Inv} = \frac{P_{Load_{Peak}}}{\eta_{inv}} \tag{6}$$

**2.3.4 Diesel generator (DGen) technical model.** The diesel generator (DGen) is used as a backup power source. It is triggered only when PV and BSS are not able to produce sufficient power during load shedding duration. The output power of the DGen is given by [32], i.e.

$$P_{DGen}(t) = P_n \times N_{DGen} \times \eta_{DGen} \tag{7}$$

In Eq (7), $P_n$ is the rated power of the generator (provided by the manufacturer in kW); $\eta_{DGen}$ is the efficiency of and $N_{DGen}$ represents the total number of generators installed in the system. The sizing of the DGen is dictated by the peak demand; this is to ensure the availability of the supply is guaranteed. Although capital investment for the generators has to be fulfilled, the operational cost can be significantly reduced by the HES.

**2.3.5 Utility grid technical model.** The grid maintains the balance between power supply and the consumption. It is modelled according to the load requirement, expressed as

$$P\_Grid(t) = R_{Grid}(t) \times P\_Load(t) \tag{8}$$

The $R_{Grid}$ is the reserve margin coefficient. The value for the $R_{Grid}$ is assumed to be 1.2, considering the strained energy conditions and the social perspective of Pakistan. However, during load shedding, the power output of the grid becomes

$$P\_Grid(t) = \begin{cases} B_{Grid}(t) \times P\_Load(t), & B_{Grid}(t) > 0 \\ 0, & B_{Grid}(t) = 0 \end{cases} \tag{9}$$

Where $B_{Grid}(t)$ is a binary variable that describes the state of the grid at a specific hour. The value for $B_{Grid}$ is determined according to actual load shedding schedule imposed by the grid operator.

## 2.4 Economic modelling

The economic model assesses the effective energy cost derived from PV, BSS and DGen and inverter, based on the levelized cost of electricity (LCOE) [33, 34]. The LCOE is defined as the total costs (in $) of the system over its warranted lifetime divided by the total energy

production (in kWh) across the same period [35], i.e.

$$LCOE = \frac{Total\ System\ Cost}{Total\ Energy\ Production} = \sum_{n=1}^{N} \frac{Cost_{System}/(1+r)^n}{Energy_{System}/(1+r)^n} \quad \left(\frac{\$}{kWh}\right) \tag{10}$$

Here, $N$ represents the project lifetime while $r$ is the discount rate. Moreover, the total system cost ($Cost_{system}$) comprises three main components: 1) the initial cost (investments) of the assets ($IC_{System}$), 2) the operating cost (insurance, running, repairs) ($OC_{System}$) over the project lifetime and 3) the replacement cost of defected assets ($RC_{System}$). Mathematically,

$$Cost_{System} = IC_{System} + OC_{System} + RC_{System} \tag{11}$$

For the HES, the system costs $Costs_{HES}$ is the sum of the total costs of the PV, BSS, DGen and the inverter. This can be written as:

$$Cost_{HES} = Cost_{PV} + Cost_{BSS} + Cost_{Inv} + Cost_{DGen} \tag{12}$$

Consequently, the total energy $Energy_{HES}$ for HES is energy generated by PV and DGen. i.e.

$$Energy_{HES} = Energy_{PV} + Energy_{DGen} \tag{13}$$

**2.4.1 PV cost model.** Since the lifetime of PV is considered to be same as the project lifetime, the replacement cost is zero. Thus, the total cost for PV is

$$Cost_{PV} = IC_{PV} + \frac{\sum_{n=1}^{N} OC_{PV}}{(1+r)^n} \tag{14}$$

where $r$ is the discount rate (assumed at 9%), $N$ represents the project life time (20 years) while $n = 1$ refers to the first year (beginning) of the project [36]. The annual energy output from the PV system is estimated by a commonly used equation [37], i.e.

$$Energy_{PV} = A \times r \times H \times PR \tag{15}$$

Here $A$ is the total area of solar panel (in m²), $r$ is the total solar panel yield (in %), $H$ represents the annual average solar radiation on the tilted panels (excluding shadings) (in W/m2), and $PR$ is the performance ratio that is the coefficient for losses (in %). The default value of 0.75 for $PR$ is considered. Furthermore, the quality of the PV system is quantified using the performance index (PI) [38]. After an exposure time of one year, the mean value of PI is typically in the range of 85–86%, which implies the performance of real PV system degrades over the years. To adjust the difference, the discounted energy and the degradation rate is included [36]. Thus the $Energy_{PV}$ becomes

$$Energy_{PV}(t) = \sum_{n=1}^{N} \frac{Energy_{PV}(1 - DEG_{PV})^n}{(1+r)^n} \tag{16}$$

The $DEG_{PV}$ represents the degradation rate of PV and considered 2.5% for the first year while 0.7% for the subsequent years. The relevant cost of PV system can be found in Table 1.

**2.4.2. BSS cost model.** For batteries, the operating cost is negligible. However, the lifetime of typical Li-ion batteries is 5 years; thus, they need to be replaced three times during the project lifetime. Assuming the replacement equals initial cost, the total costs of the BSS is calculated

**Table 1. Technical specifications of HES components.**

| Component | Parameter | Variable | Values | Units |
|---|---|---|---|---|
| PV | Rated power capacity (per module) | $C_{PV}$ | 325 | W |
| | Module efficiency | r | 17.0 | % |
| | Performance ratio | PR | 0.75 | - |
| | Initial (capital) cost | $IC_{PV}$ | 305 | $/kW |
| | Operating cost (yearly) | $OC_{PV}$ | 3.05 | $/Kw |
| | Expected lifetime | $Life_{PV}$ | 20 | Years |
| BSS | Rated capacity | $C_{Bat}$ | 1800 | Wh |
| | Charging/discharging efficiency | $\eta_{Bat}$ | 100 | % |
| | Initial (capital) cost | $IC_{Bat}$ | 120 | $/kW |
| | Operating cost (yearly) | $OC_{Bat}$ | 120 | $/kW |
| | Expected lifetime | $Life_{Bat}$ | 5 | Years |
| DGen | Rated power | $P_{DGen}$ | 30 | kW |
| | Generator efficiency | $\eta_{DGen}$ | 85.0 | % |
| | Power factor | PF | 0.8 | - |
| | Initial (capital) cost | $IC_{DGen}$ | 100 | $/kW |
| | Operating cost (yearly) | $OC_{DGen}$ | 0.064 | $/Hour |
| | Fuel cost | $FC_{DGen}$ | 0.812 | $/Liter |
| | Expected lifetime | $Life_{DGen}$ | 15000 | Hours |
| Inverter | Rated power | $P_{inv}$ | 30 | kW |
| | Inverter efficiency | $\eta_{inv}$ | 95 | % |
| | Initial (capital) cost | $IC_{Inv}$ | 1669 | $ |
| | Expected lifetime | $Life_{Inv}$ | 20 | Years |

as

$$Costs_{BSS} = IC_{BSS} + \frac{\sum_{n=6,11,16} RC_{BSS}}{(1+r)^n} \tag{17}$$

Batteries do not generate their own energy; they are charged by PV and the grid. Thus, the BSS energy is based on the contribution from PV and the grid, i.e.

$$Energy_{BSS}(t) = \sum_{n=1}^{N}\sum_{t=1}^{8784} PV\_ch\_BSS(t) + \sum_{t=1}^{8784} Grid\_ch\_BSS(t) \tag{18}$$

**2.4.3 DGen cost model.** Besides the capital investment, the total costs of DGen also includes the fuel consumption. The latter contributes a considerable portion of the generation cost. The fuel cost is given by

$$Fuel_{Consume}(t) = \begin{cases} C_1 \times P_{DGen,rated} + C_2 \times P_{DGen}(t), & P_{DGen}(t) > 0 \\ 0, & P_{DGen}(t) = 0 \end{cases} \tag{19}$$

The $P_{DGen,rated}$ and $P_{DGen}$ represent the rated power and the generated power of DGen (in kW), respectively. Meanwhile $C_1$ and $C_2$ refer to the coefficients of fuel consumption curve.

Their values are 0.246 and 0.085 respectively [39]

$$FC_{DGen}(t) = \begin{cases} Fuel_{Consume}(t) \times f_P, & Fuel_{Consume}(t) > 0 \\ 0, & Fuel_{Consume}(t) = 0 \end{cases} \qquad (20)$$

The $f_P$ is the fuel price (in \$/L). Accordingly, the yearly lifetime of DGen ($Life_{\text{DGen,year}}$) can be computed using Eq (21). The $Life_{\text{DGen,year}}$ is required to find the replacement cost for DGen.

$$Life_{DGen,year} = \frac{Life_{DGen,hour}}{WH_{DGen,year}} \qquad (21)$$

The total costs and the energy of the DGen can be found using

$$Costs_{DGen} = IC_{DGen} + \frac{\sum_{n=1}^{N} OC_{DGen}}{(1+r)^n} + \frac{\sum_{n=1}^{N} RC_{DGen}}{(1+r)^n} + \frac{\sum_{n=1}^{N} FC_{DGen}}{(1+r)^n} \qquad (22)$$

$$Energy_{DGen}(t) = \sum_{n=1}^{N} \frac{\sum_{t=1}^{8784} Energy_{DGen}}{(1+r)^n} \qquad (23)$$

The data required for the economic models of HES components is provided in Table 1. Further, the reliability of the proposed model is analysed as follows.

## 2.5 Reliability analysis

The system reliability is measured in terms of its ability to supply the required power reliably [40]. The utilities have opted various approaches for reliability evaluation of hybrid system; among the most common are loss of load expectation (LOLE), loss of load probability (LOLP) and the loss of power supply probability (LPSP). The reliability analysis of the proposed HES is based on the LPSP. It is defined as the probability of power system failure to meet the required load demand, i.e.

$$LPSP = \frac{T_{ft}}{T} \qquad (24)$$

where $T$ represents the complete time horizon of the analysis while the $T_{ft}$ shows the power failure time, i.e., the time when the load demand is not satisfied. Mathematically this is written as

$$T_{ft} = \sum_{t=1}^{T} \phi(t) \Delta t \qquad (25)$$

$$\phi(t) = \begin{cases} 1 & P_{RES}(t) < P_{Load}(t) \\ 0 & Otherwise \end{cases} \qquad (26)$$

The $\Delta t$ is the sampling time (one hour) while $P_{RES}(t)$ is the power supplied by PV and BSS. The value of LPSP ranges between 0 to 1. Zero LPSP means the load demand is fully satisfied; on the other hand, if LPSP is unity, the demand is not satisfied at all.

## 2.6 Case study

The viability of the proposed HES is analysed for a small residential community situated in the city of Quetta, Pakistan. Since the local grid cannot satisfy the demand of the community for

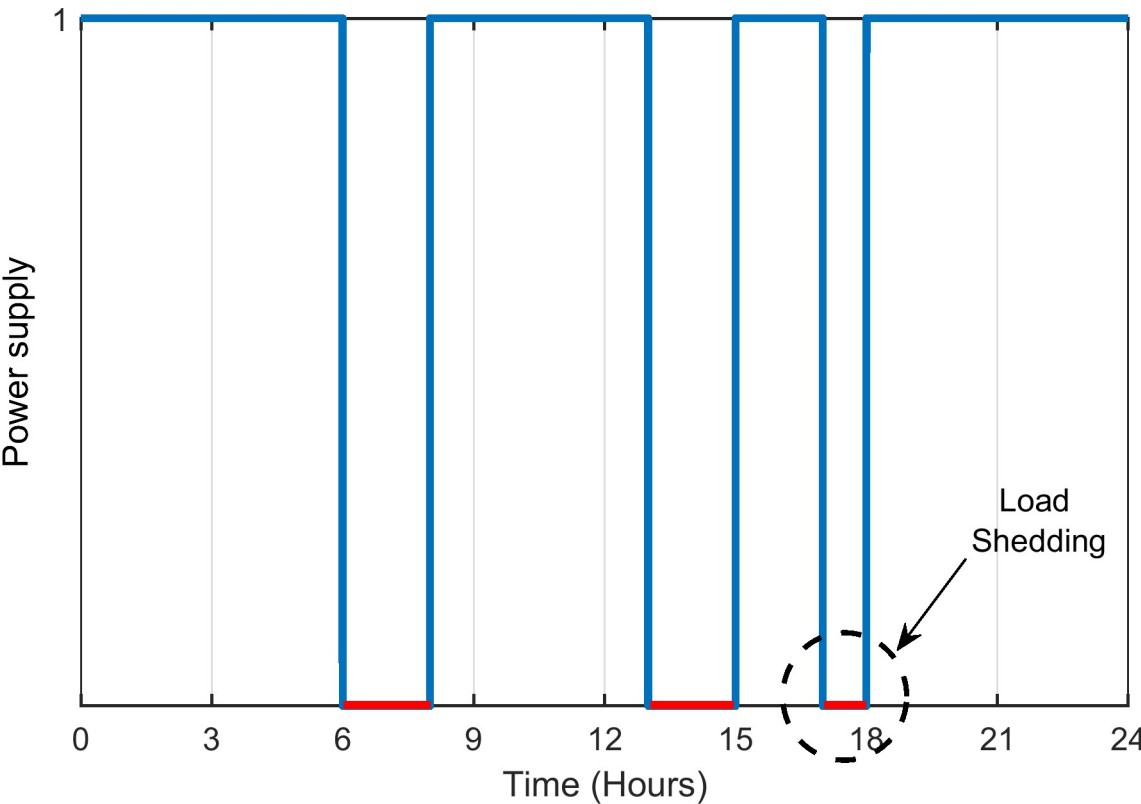

**Fig 4. Power grid supply during summer.**

24 hours, the load shedding is scheduled. Its occurrence is known beforehand, as presented in Fig 4. The given profile is based on the real time load-shedding scenario for a typical summer day (14th June 2020) [41]. The grid profile is plotted in the form of a unit-step function; the red curve represents the unavailability of supply, which is manifested by the load shedding occurrence. For this particular day, the total load shedding duration is 5 hours. However, the schedule differs over the course of the year.

The load demand is shown in Fig 5. In this case study, power demand for 40 houses with limited public facilities is considered [42]. Two load profiles are considered: one for the summer (April to October) and the other for the winter season (November to March). The daily energy consumption for the summer and winter is 345 and 309 kWh, respectively, while the peak power usage is at 26.66 and 23.91 kW. The region surrounding Quetta has the great potential for renewables, particularly solar. It is characterised by high solar irradiance and moderate temperature, which make this region suitable for PV generation. Figs 6 and 7 demonstrate the annual solar radiation and ambient temperature of Quetta respectively, logged from August 2019-August 2020 [26].

## 2.7 The energy management scheme (EMS)

To ensure uninterrupted supply at an optimal cost, EMS controls the HES operation by two operating modes i.e., *Grid mode* and the *Islanded mode*. Former is activated when the grid is supplying power and later is triggered during the absence of the grid power. The operation is executed by a controller that automatically controls the system without the physical presence of the human operator. The modes are explained below.

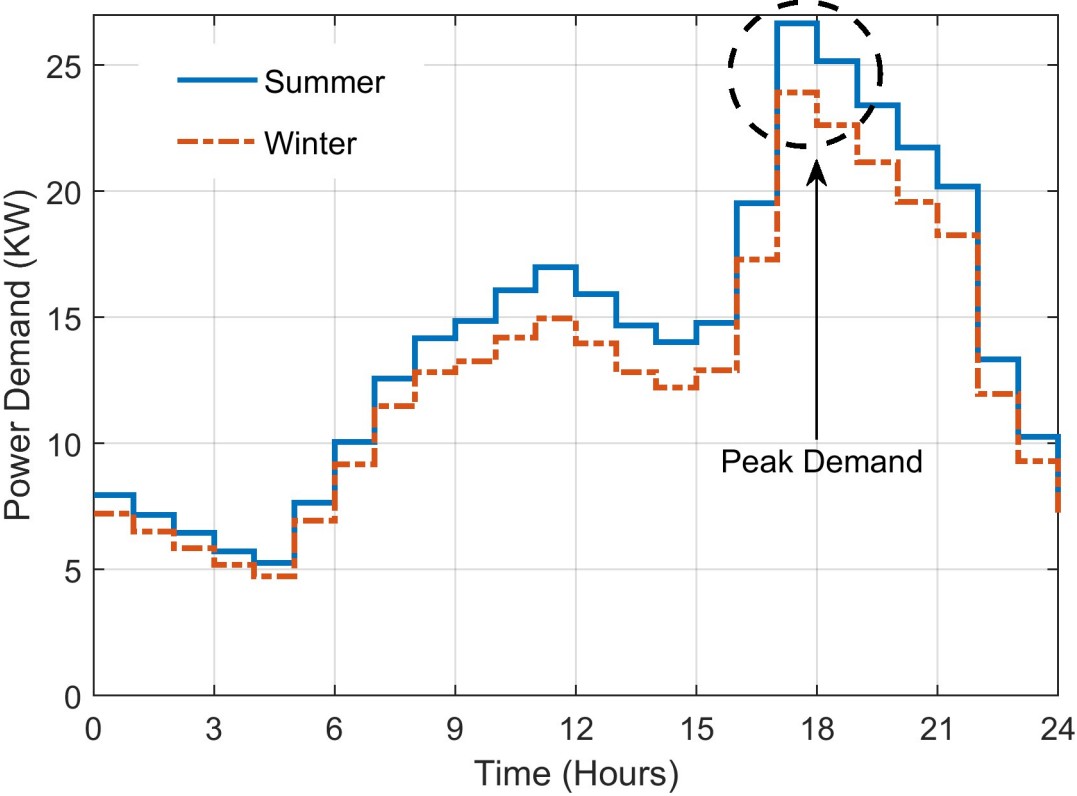

**Fig 5. Community load demand on hourly basis.**

- *Grid mode*: the grid is assumed to have sufficient power to fulfil the load demand. There are several possible scenarios: if grid has surplus power, it is used to charge the BSS. Since the research considers the TOU tariff policy, the BSS charging takes place during the off-peak hours only. Meanwhile, if PV starts producing power, it will be prioritized to charge the BSS.

- *Islanded mode*: activated when power from utility grid is not available, i.e., during the occurrence of load-shedding. The grid is powered by PV, BSS, and DGen. For maximum economic benefit, the priority is given to extract the PV energy. However, since the PV is intermittent, the BSS is utilized to complement the energy deficit. Furthermore, there are two possible scenarios:

   Scenario 1: $P_{PV}(t) > P_{Load}(t)$; the extra PV power is used to charge BSS provided the $SOC < SOC_U$.
   Scenario 2: $P_{PV}(t) < P_{Load}(t)$; the load requirement is met by BSS, provided the $SOC > SOC_L$.
   On the other hand, DGen is triggered when the grid, PV and BSS are no longer able to meet the load demand. The DGen only covers the power deficit to ensure supply availability during load shedding.
   Fig 8 shows the flowchart of the overall HES operation. For each simulation time step (hourly), the input data (irradiance, temperature, $SOC$, $COE_{Grid}$, $LCOE$, load demand, shedding schedule) are read. Then, the values of power ($P_{PV}$, $P_{BSS}$, $P_{Grid}$, $P_{DGen}$) are updated. Based on the input data, operating scenarios are created and the appropriate modes of the HES are determined. Under these operating modes, different (single or multiple) functions are

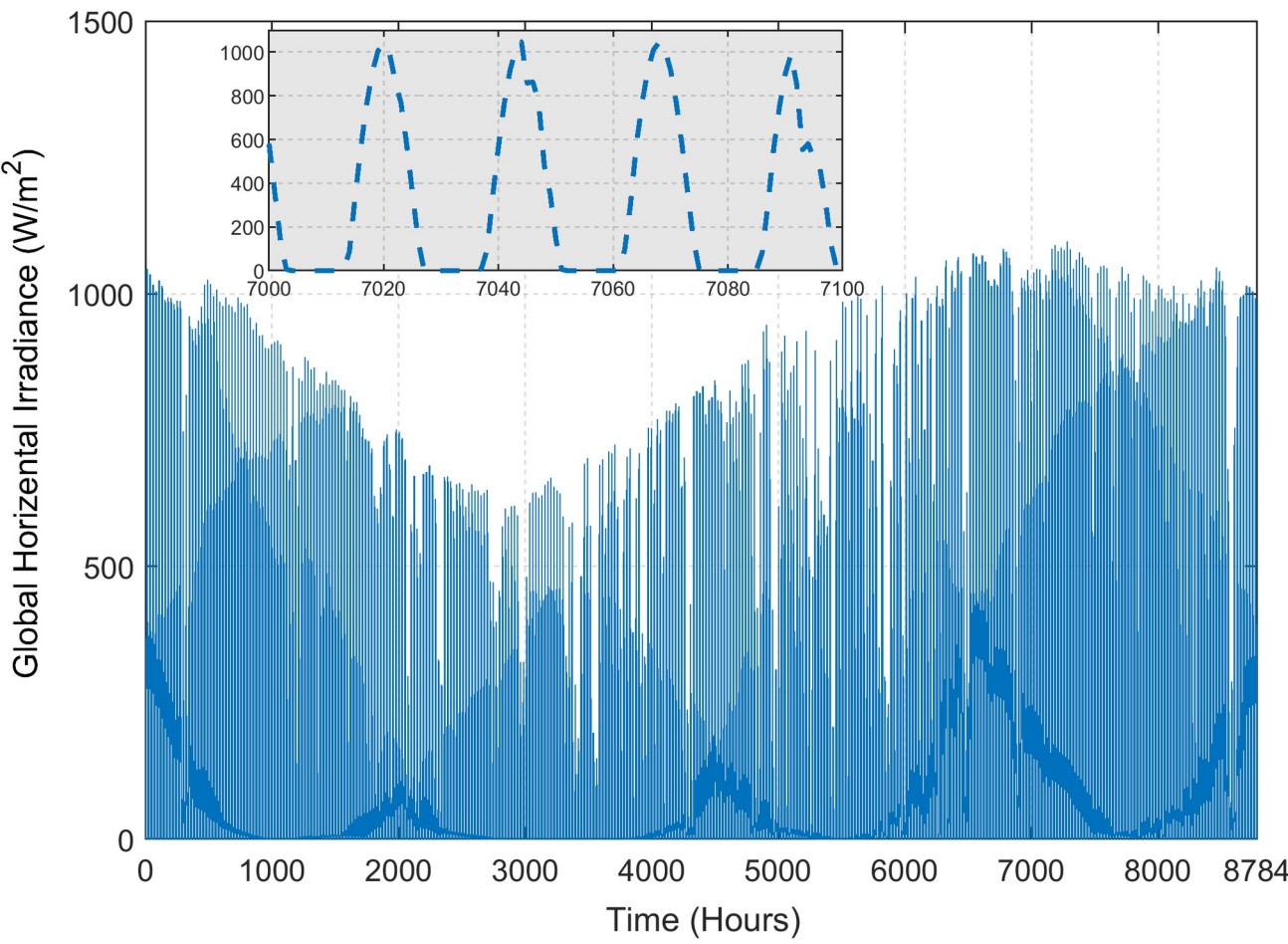

**Fig 6. Annual solar radiation of Quetta.**

performed to achieve the required objectives of EMS. These functions maintain the record of events happening under the active operating modes. For instance, the PV energy transactions are tracked through the *PV_supp_Load*, *PV_ch_BSS* and *PV_feed_Grid* functions. The *BSS_dis* manages the BSS discharging operation during the RES mode. Accordingly, the grid mode operations are controlled with the help of *Grid_supp_Load* and *Grid_ch_BSS* while the DGen power supply is performed by the *DGen_supp_Load* in DGen mode. After every iteration, the operational cost for each mode (based on the electricity prices and the energy contribution) are calculated.

## 2.8 Formulation of objective function

The main objectives of HES are to ensure supply availability and to minimize the *LCOE*. This is achieved by determining the minimum number of modules ($N_{PV}$) and BSS batteries ($N_{Bat}$) that can fulfil the load demand under specific operational constraints. The relationship between *LCOE* and component sizing can be expressed as

$$\min f(n_p) = \min \left( \sum_{t=1}^{T} LCOE(t) \right) \tag{27}$$

*T* is the duration of the load shedding intervals (in hours). Variable $n_p$ is the vector of sizing variables; in this case, $n_p = \{N_{PV}, N_{Bat}\}$. When these values are optimized, the LCOE is expected

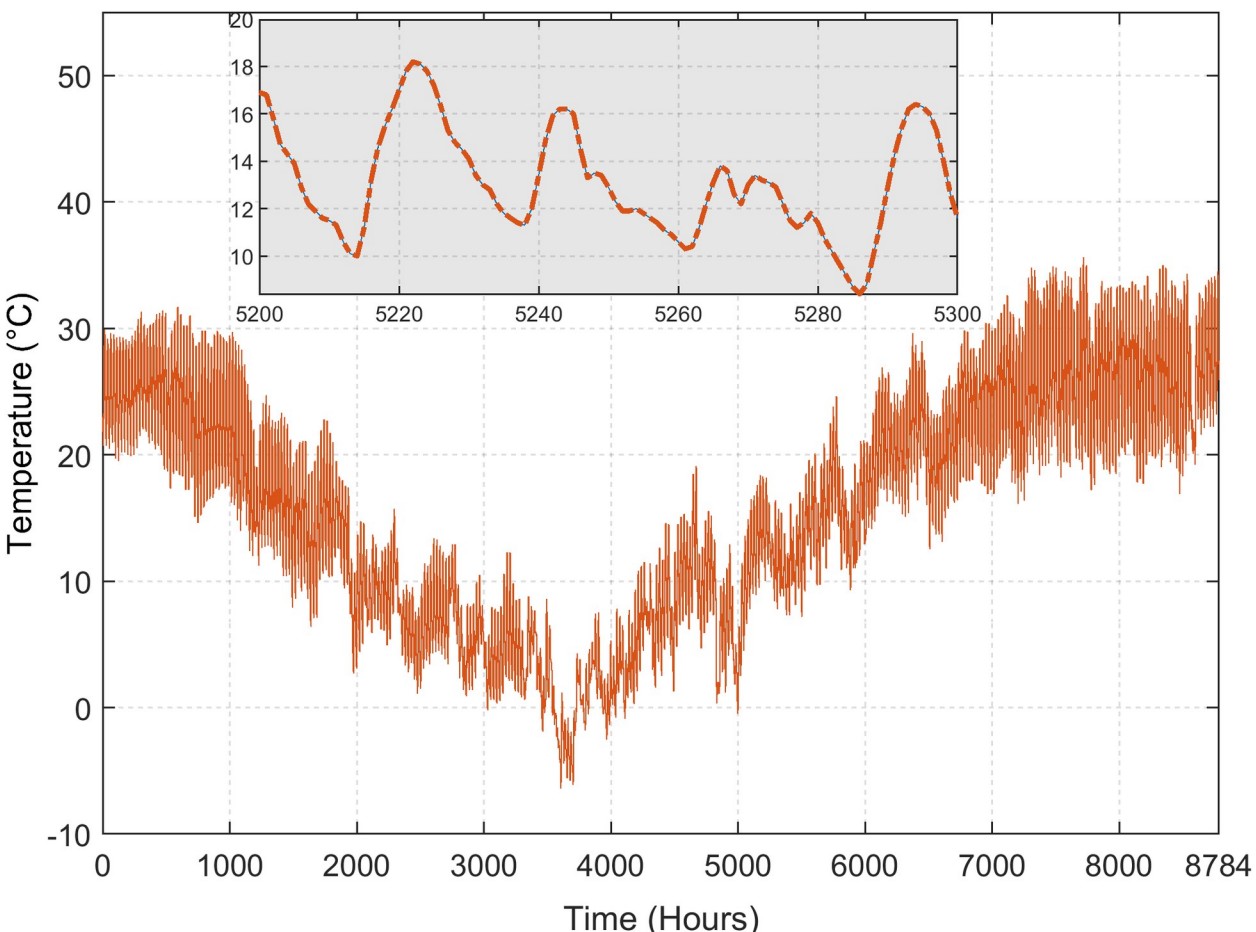

**Fig 7. Annual average temperature of Quetta.**

to be at its lowest value. The power generated from the RES can be calculated as:

$$P_{RES} = N_{PV} \times C_{PV} + N_{Bat} \times C_{Bat} \tag{28}$$

Where $C_{PV}$ and $C_{Bat}$ are the rated power capacities of one PV module and battery unit, respectively. The values for $C_{PV}$ and $C_{Bat}$ are given in Table 1.

**2.8.1 Operational constraints.** The objective function is subject to the fulfilment of following constraints.

1. The LPSP is set to zero. This implies that hourly load demand must be satisfied i.e. only those combination of $N_{PV}, N_{Bat}$ should be considered that can fulfil the load demand.

2. Renewable energy contribution (**RC**): The *RC* is the portion of energy generated from renewable energy. The *RC* is set to ≤ 20% as per Pakistan government allowed quota. The *RC* can be determined using

$$RC = \frac{\sum_{t=1}^{8784} PV\_feed\_Grid(t)}{\sum_{t=1}^{T}(PV\_feed\_Grid(t) + P_{Grid}(t))} \tag{29}$$

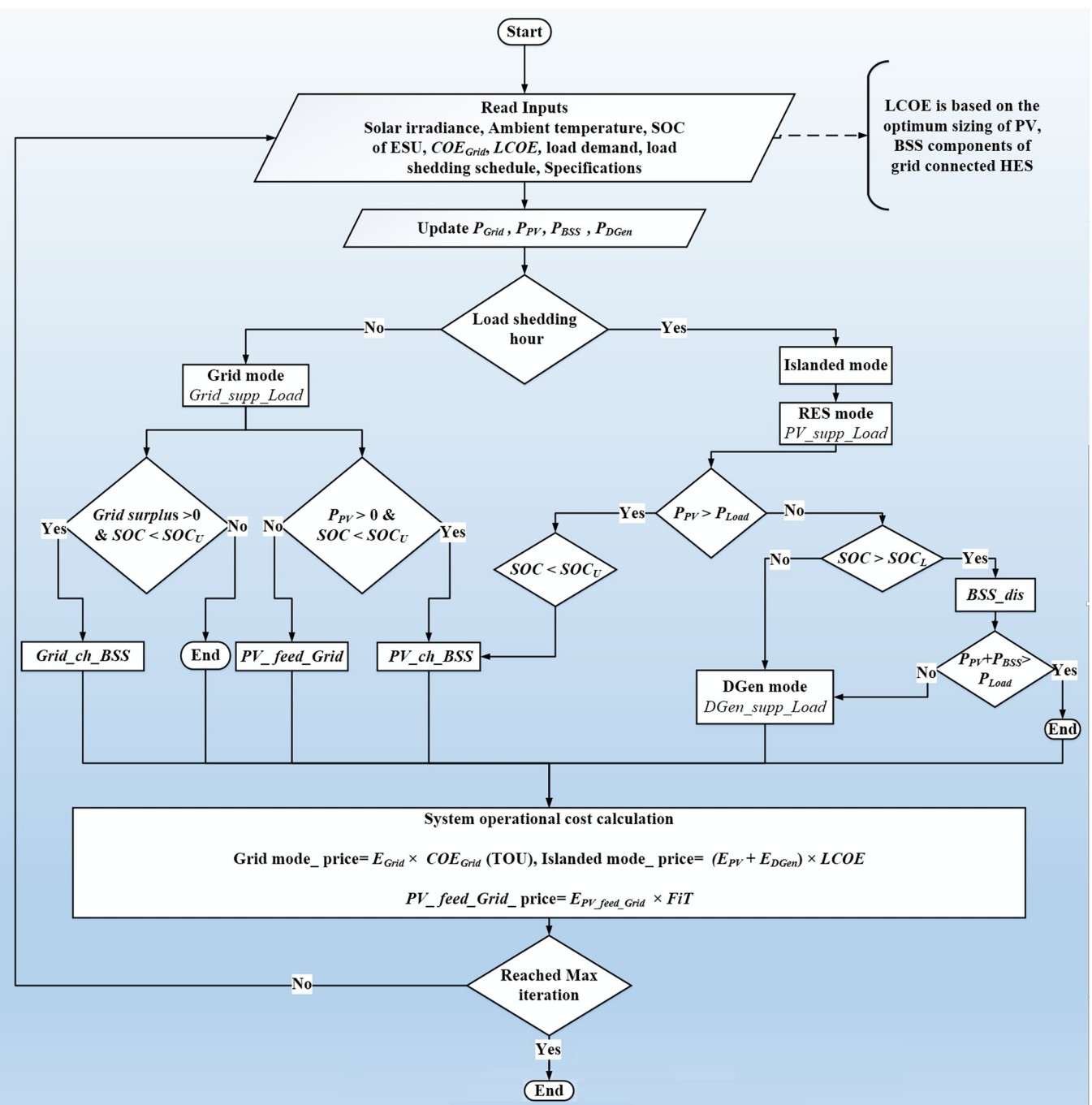

**Fig 8. The flow chart of EMS algorithm for the HES.**

3. Moreover, the following inequality constraints for $N_{PV}$ and $N_{Bat}$ should be satisfied.

$$N_{PV} = \text{integer}, \ N_{PV}^{\min} \leq N_{PV} \leq N_{PV}^{\max} \tag{30}$$

$$N_{Bat=}\text{integer}, \ N_{Bat}^{\min} \leq N_{Bat} \leq N_{Bat}^{\max} \tag{31}$$

The $N_{PV}^{\min}$ and $N_{PV}^{\max}$ are minimum and maximum permissible number of PV modules, respectively. Likewise, $N_{Bat}^{\min}$ and $N_{Bat}^{\max}$ represent the minimum and maximum number of permissible battery units, respectively. The limits are considered on hit-and-trial basis to reduce the convergence time of PSO.

**2.8.2 Optimal sizing of HES components using Particle swarm optimization.** Particle swarm optimization (PSO) is a well renowned metaheuristic optimization algorithm employed to solve various problems in engineering applications [43–45]. The algorithm mimics the social conduct of swarm of birds when they search for food in nature. The swarm behaviour of PSO is modelled mathematically as

$$x_i^{k+1} = x_i^k + v_i^{k+1} \tag{32}$$

Where $x_i$ is the position of *ith* particle in the swarm while $v_i$ is the velocity component that represents the step size. The velocity can be calculated as

$$v_i^{k+1} = wv_i^k + c_1\,r_1(P_{best} - x_i^k) + c_2\,r_2(G_{best} - x_i^k) \tag{33}$$

In Eq 33, $w$ is inertia weight while $c_1$, $c_2$ are the acceleration coefficients. The $r_1$ and $r_2$ are random numbers and their value lies between (0–1). The $P_{best}$ represents the personal best position of the *ith* particle and the $G_{best}$ represents the global best position of all particles. Fig 9 illustrates the general idea of particle movement in the optimization process.

For the proposed system, the PSO is utilized to solve the sizing problem according to [46]. Different parameters of PSO used in the simulation are given in Table 2 [47]. The PSO based optimal sizing of PV and BSS components can be summarized in the following steps.

1. An initial population $n_p$ (different combinations of $N_{PV}$ and $N_{Bat}$) is randomly generated in the search space. However, only those combinations of particles ($N_{PV}$ and $N_{Bat}$) are selected that satisfy the aforementioned constraints.

2. The initial velocity (of each particles) is randomly generated using Eq 33.

3. The simulation of the HES under the control of EMS is executed; the value of LCOE for each respective combination of $N_{PV}$ and $N_{Bat}$ is calculated.

4. The initial position of each particle is considered as its personal best, i.e. $P_{best}$, while the combination of $N_{PV}$ and $N_{Bat}$ that gives minimum value of LCOE among the population is chosen as global best, i.e. $G_{best}$.

5. Particles (i.e. $N_{PV}$, $N_{Bat}$) update their positions based on Eqs 32 and 33. However, only eligible combinations of particles (that satisfy the aforementioned constraints) are selected.

6. Simulation is run again and the objective function value against new combination of $N_{PV}$ and $N_{Bat}$ is calculated.

7. Values of $P_{best}$ and $G_{best}$ are updated.

8. The stopping criteria is based on the maximum number of iterations set by the user. If it is satisfied, the program is terminated. Lest, steps (5) to (8) are repeated.

9. From the obtained population, the combination ($N_{PV}$, $N_{Bat}$) that gives the lowest value of LCOE is selected as the optimal solution.

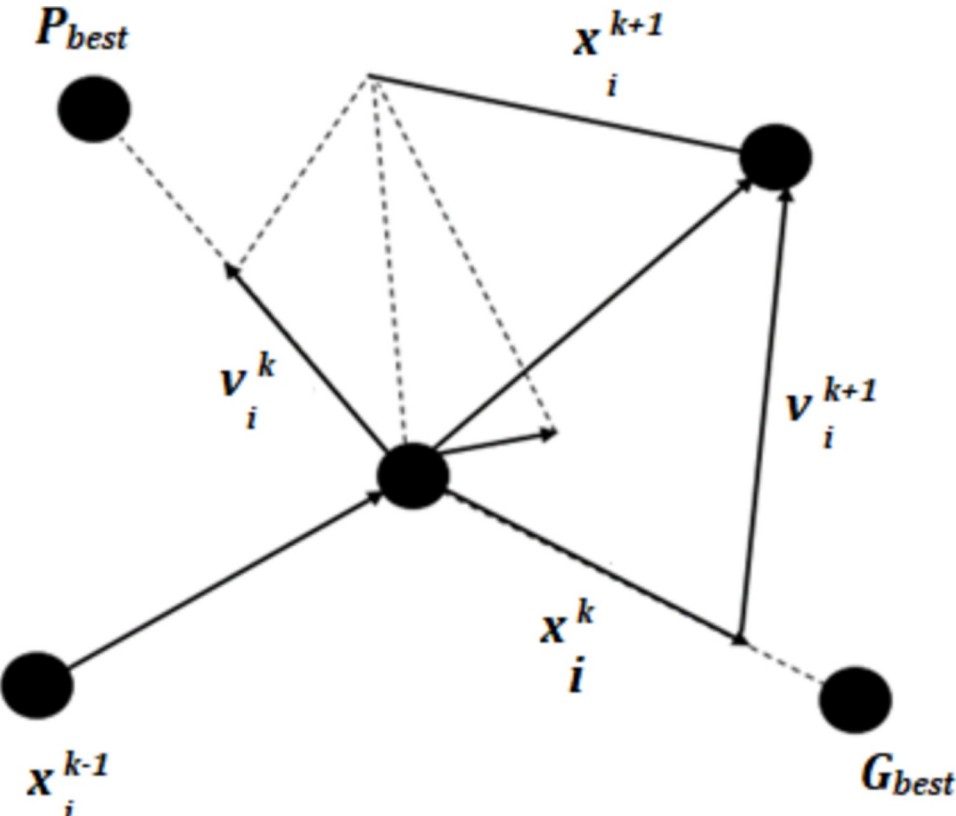

**Fig 9. Movement of particles in optimization process.**

## 3. Results and discussions

The HES is simulated using the MATLAB-Simulink environment. The simulations are performed using real data-set of load shedding and irradiance profile, published in [21, 42]. To observe the resiliency of HES in providing uninterrupted supply, two different scenarios are evaluated: one for the summer day and other for the winter day. It must be noted, however, the resiliency is not defined in the same manner as the power system resiliency (which is an index); rather it is an indicator to show that the system is able to fulfil the demand under varying operating conditions. Furthermore, the yearly performance of the system is evaluated. This is done by running the simulation of the system using the meteorological data and load shedding schedule of the entire year. For benchmarking, the HES performance is compared with conventional load shedding mitigation methods (UPS, generator) that does not incorporate the PV. The results showing HES operation during summer and winter days are presented in

**Table 2. PSO parameters.**

| Variable | Symbol | Value |
|---|---|---|
| Population size | $n_p$ | 10 |
| Constant acceleration factors | $c_1, c_2$ | 2 |
| Max inertia weight | $w_{max}$ | 0.9 |
| Min inertia weight | $w_{min}$ | 0.4 |
| Max iterations | $max_{iter}$ | 100 |

Figs 11 and 14, respectively. Each hourly profile is summarized in four plots: (a) status of PV energy and its utilization during HES operation; (b) energy transfer related to BSS charging and discharging process; (c) transactions of energy performed under different sources of HES and (d) resiliency of HES in the varying load conditions.

### 3.1 Scenario1: Typical sunny summer day

Fig 10 shows the global tilted irradiance profile of a specific summer day (14 June 2020). Since the cloudiness has a strong impact on irradiance that hits the surface of the module, the cloud opacity is plotted too. The low opacity indicates clear sky conditions and high incident of solar radiation [48]. During much of the daytime, 7:00 to 17:00 hours, an almost constant high availability of solar irradiance can be observed. This shows the normal conditions of the weather.

The load shedding is imposed for five hours in three separate trenches: 6–8, 13–15 and 17–18 hours. The operation of the various HES components is shown in Fig 11. In plot (a), the PV energy utilization is shown. Besides charging the BSS (*PV_ch_BSS*) and supplying the load (*PV_supp_Load*), the surplus power from the PV is sold (through the net metering option) to the grid (*PV_feed_Grid*). The net metering scheme provides attractive incentive in terms of economic benefits to the system owner. The charging/discharging profile of the BSS is shown in plot (b). As expected, the BSS charges from the grid (*Grid_ch_BSS*) during the off-peak hours. It is also charged from the surplus obtained from the PV power (*PV_ch_BSS*). The BSS discharges its energy during the load during shedding intervals (*BSS_dis*). The negative value of *BSS_dis* indicates discharging. Accordingly, the SOC variations can be seen consistent with the charging and discharging trend of the BSS. At any instant, the energy delivered by the BSS can be written as.

$$P_{BSS}(t) = \frac{P_{Load}(t)}{\eta_{inv}} - P_{PV}(t) \qquad (34)$$

It should be noted that Eq (34) may result in $P_{BSS}(t) < 0$ due to the intermittent behaviour of PV. This situation represents the amount of PV energy being absorbed by the BSS, as

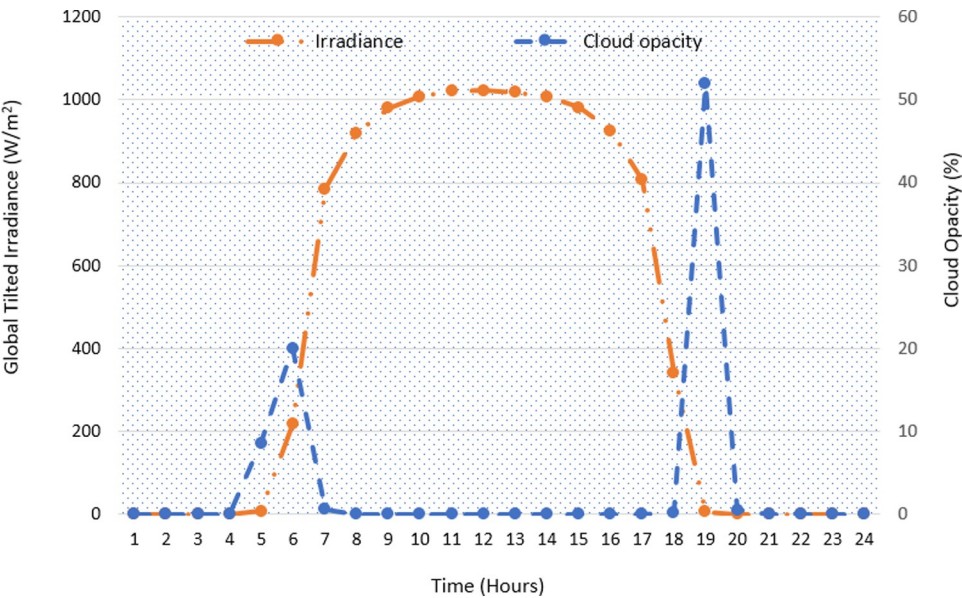

**Fig 10. The global tilted irradiance and the cloud opacity (on 14 June 2020).**

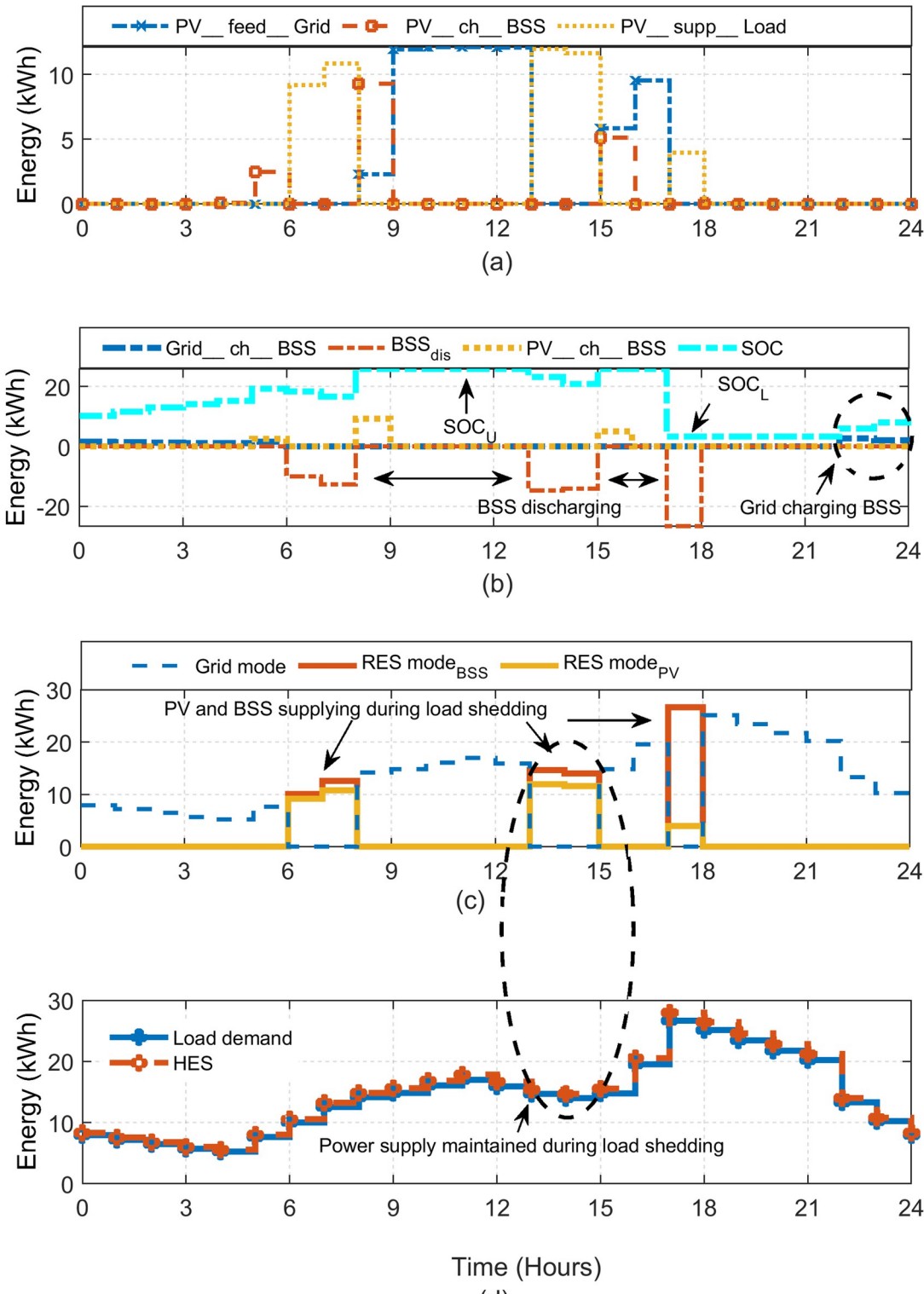

**Fig 11.** The HES performance during a typical summer day (a) Different modes of PV system (b) BSS charging and discharging operation with SOC variations (c) Utilization of Grid, PV and BSS (d) The resiliency of HES against variable load.

dictated by Eqs (4) and (5). Fig 11(C) shows the main energy flow by different sources of HES. Note that during the load shedding intervals, the grid is maintained by PV and/or BSS. The PV is utilized on first priority, while the BSS is used as a support in the case if the former is insufficient. As long as the coordinated energy exchanges from PV and BSS are able satisfy the load demand, the DGen is not triggered. Finally, the profile of the power supplied by the HES against the varying load demand is shown in Fig 11(D) The results show that despite the significant load shedding imposed on the grid, a stable and the uninterrupted supply is maintained for the 24-hour period. This proves that the HES is sufficiently resilient in providing continuous supply.

The system is simulated for various combination of $N_{PV}$ and $N_{Bat}$. During the simulation, following ranges are decided for the decision variables: $50 < N_{PV} < 70$ and $10 < N_{Bat} < 20$. Ten best-selected results with respect to the LCOE are given in Table 3. It can be observed that all constraints (i.e., LPSP, RC) are satisfied. The best combination (in terms of minimum LCOE) is 67 PV modules and 16 battery units. For this optimized combination, the computed value of the LCOE is 8.32 cents/kWh. This value is lower than the grid electricity price (9.3 cents/kWh) of Pakistan [49]. These results suggest that the HES is a viable solution to the load shedding problem. Besides, the utilized modelling and optimization approach provides better results than relevant studies in literature [20, 50, 51]. However, the economic feasibility of HES differs from case to case, depending on the case conditions and the size of the system. Thus, for validation, it is required to simulate different systems under similar operating conditions of HES to allow fair comparison of these systems.

Furthermore, the results show the inverse relationship between RC and LCOE. It is evident that LCOE decreases with a higher contribution of PV and vice versa. The impact of RC on the electricity price can be observed in Fig 12.

## 3.2 Scenario 2: Typical winter day

The solar irradiance and the cloud opacity for the selected winter day (7th November 2019) is shown in Fig 13. As shown, the irradiance for this day is much lower than the summer day. Further, the delayed sunrise and the early sunset in the winter season also reduce the total duration of received irradiance. Specifically, the influence of clouds on the irradiance can also be observed. The reduction in irradiance at hour 14 reflects the cloud cover.

The HES operation is illustrated in Fig 14. In this scenario, the load demand for the winter season is used while the scheduled load shedding of 4 hours is considered. Also, the same configuration ($N_{PV}$, $N_{Bat}$) is used. Since PV is not producing sufficient power, the BSS has to

**Table 3. Result of sizing problem in different combinations.**

| No. of PV Modules ($N_{PV}$) | No. of battery units ($N_{Bat}$) | LPSP | LCOE (US$ Cents/kWh) | RC (%) |
|---|---|---|---|---|
| 65 | 16 | 0 | 8.46 | 18.96 |
| 66 | 16 | 0 | 8.39 | 19.42 |
| 67 | 16 | 0 | 8.32 | 19.88 |
| 60 | 16 | 0 | 8.85 | 16.85 |
| 64 | 18 | 0 | 8.87 | 18.26 |
| 65 | 20 | 0 | 9.12 | 18.59 |
| 59 | 18 | 0 | 9.29 | 16.24 |
| 65 | 17 | 0 | 8.62 | 18.69 |
| 66 | 17 | 0 | 8.55 | 19.16 |
| 64 | 19 | 0 | 8.87 | 18.94 |

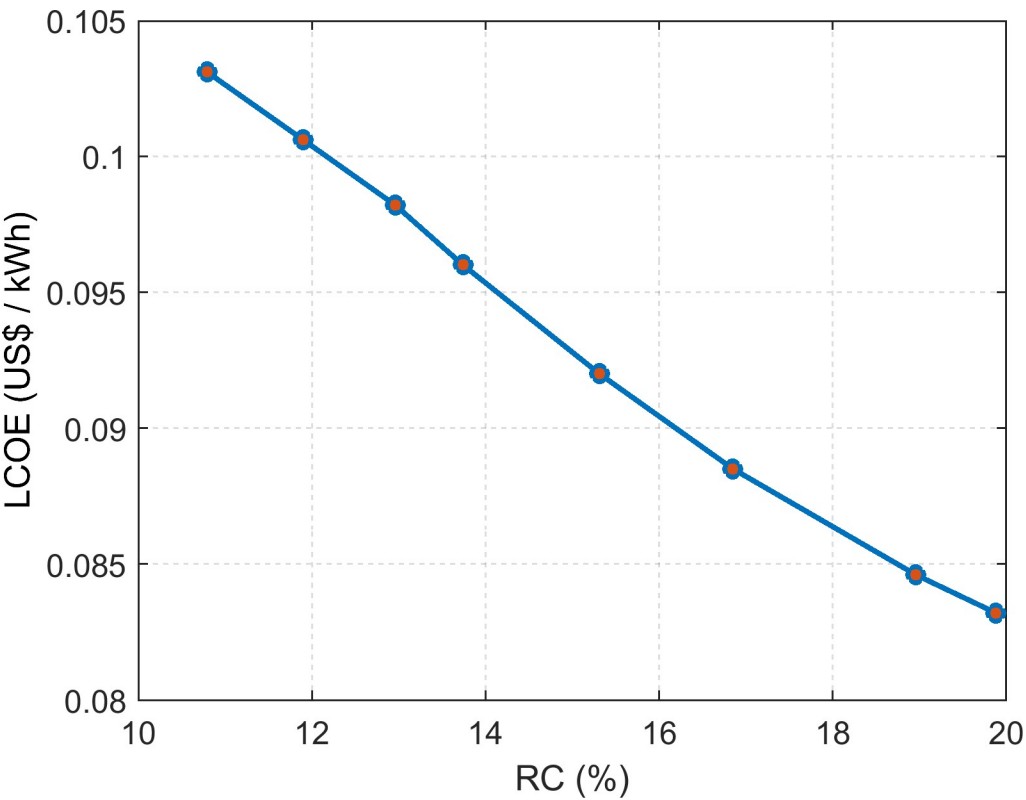

**Fig 12. The LCOE and the RC relationship for HES.**

discharge its energy to support the grid during the shedding duration, as shown in plot (a).
The BSS operation (charging and discharging) is shown in Fig 14(B). The energy transactions
and flows of different sources are demonstrated in plot (c). Accordingly, the HES operation
with variable load can be observed in plot (d). Despite the limited PV yield, the HES can satisfy
the load demand without turning on the DGen. However, the BSS reaches its maximum

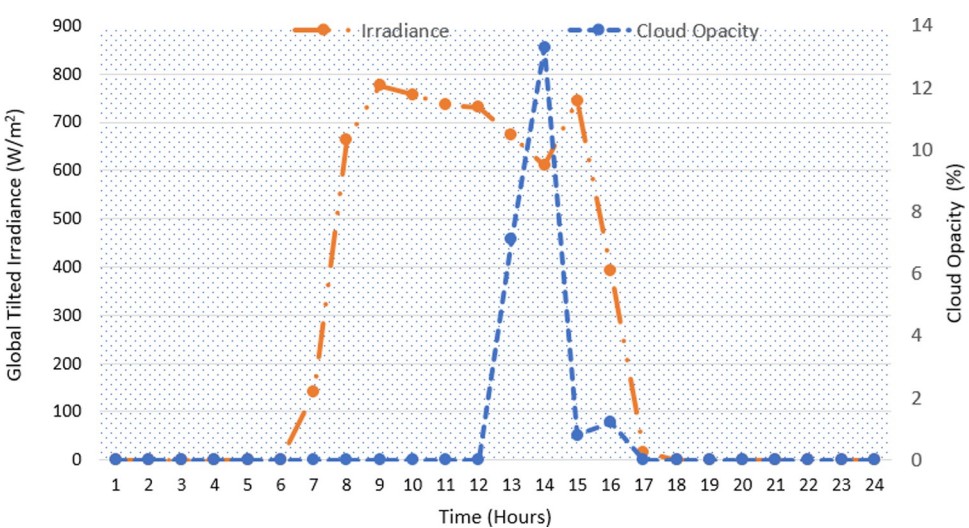

**Fig 13. The global tilted irradiance and the cloud opacity (on 7 November 2019).**

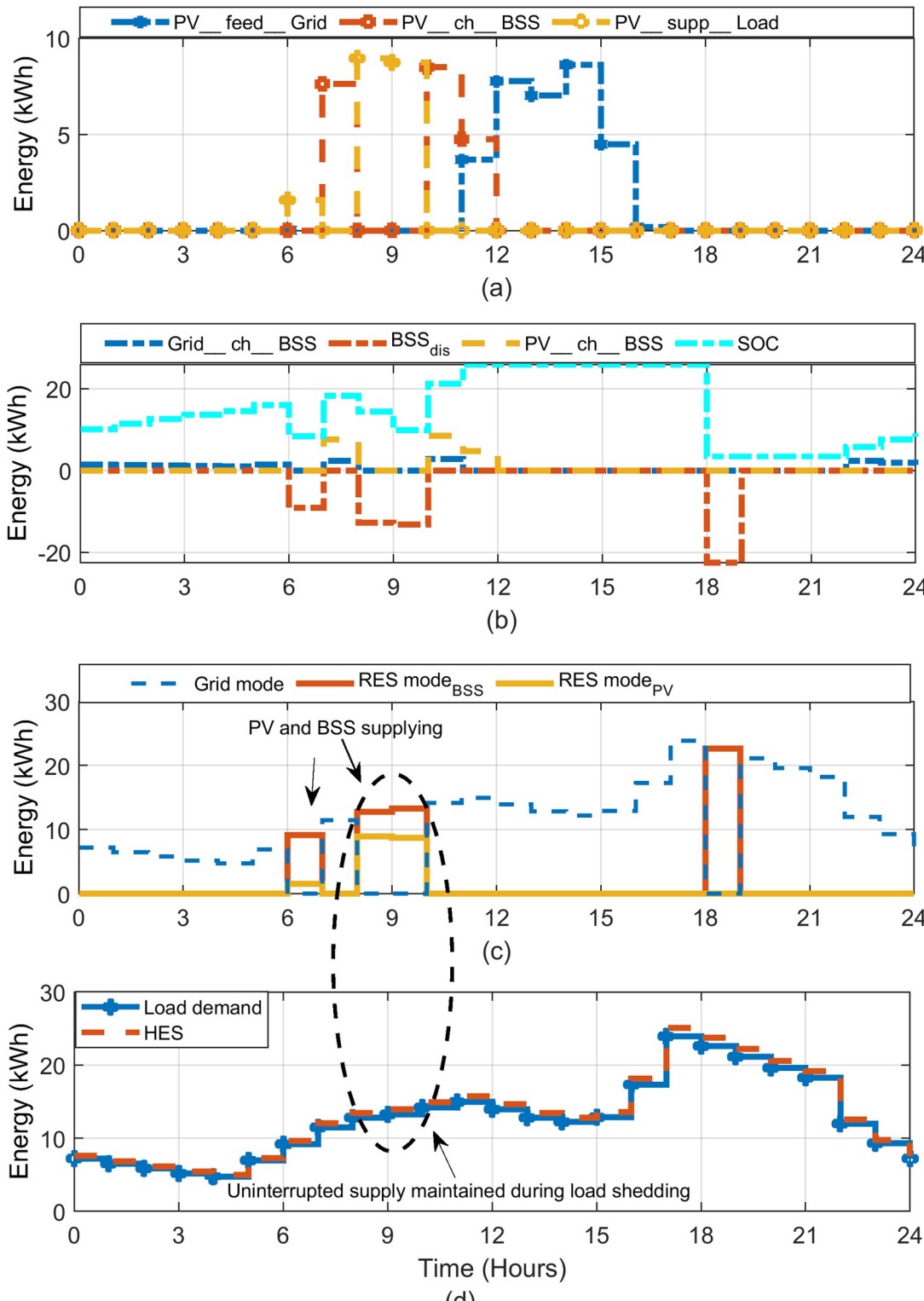

**Fig 14.** Operation of HES during winter day (7th November 2019) (a) Different modes of PV system (b) BSS charging and discharging operation with SOC variations (c) Utilization of Grid, PV and BSS (d) The resiliency of HES against variable load.

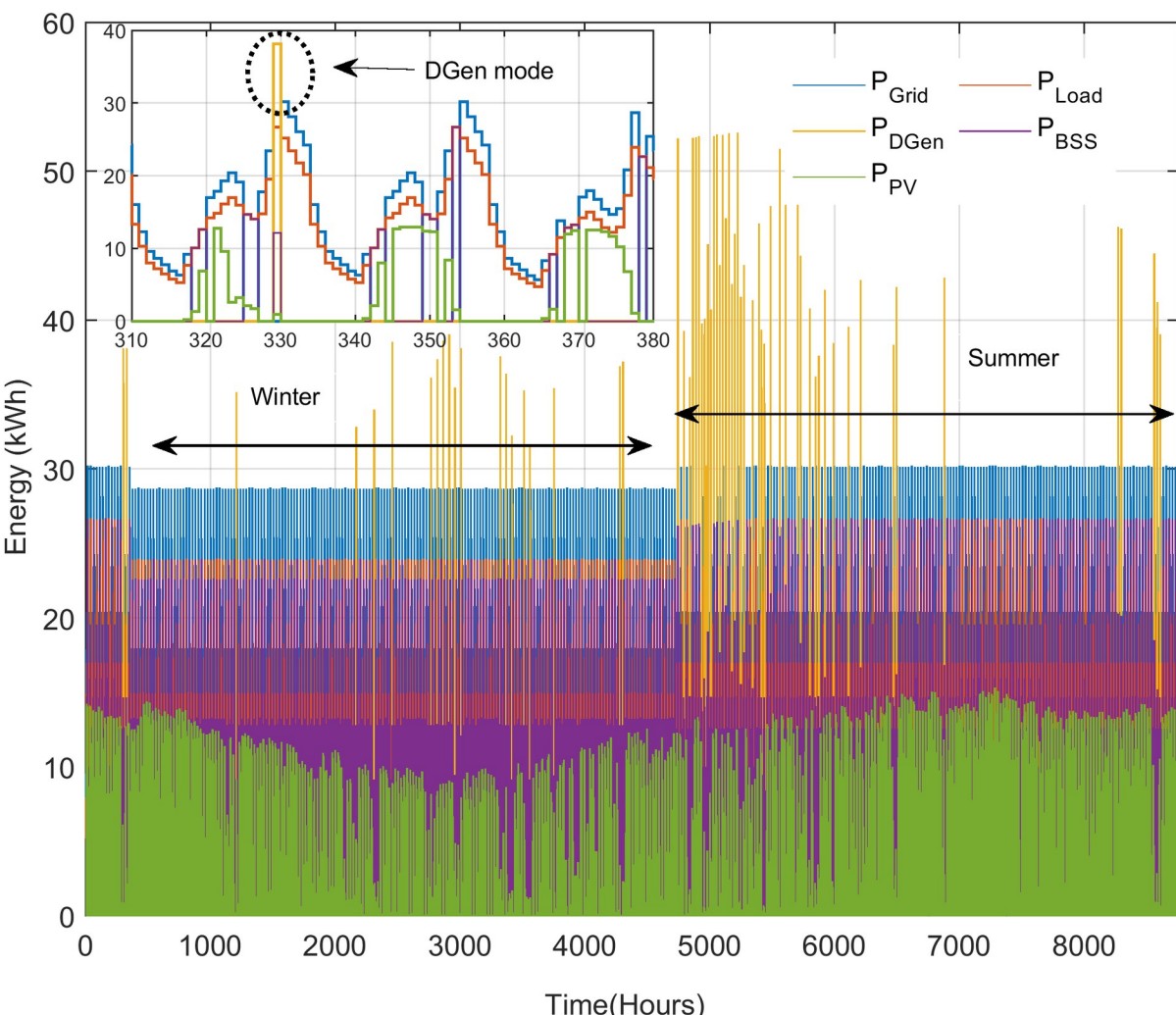

**Fig 15. The annual energy generation mix of HES and the load demand.**

allowed limit to discharge. Clearly, the system resiliency can be observed throughout the day. In addition, the continuous exploitation of PV and surplus grid power indicates the optimized utilization of the available energy. Furthermore, the HES also reduces the burden on the grid. This is because of PV support to grid operations during the daytime.

Obviously, the two scenarios above cannot be used to generalize the overall performance of the HES. These cases are meant to illustrate the operation of the HES with the EMS. For further assessment, simulation using long-term data is needed. Thus, the logged data from August 2019 to August 2020 (8,784 hours) are used, as demonstrated in Fig 15. In particular, the DGen operation is zoomed during the critical conditions; during these times, the PV and battery are not able to support the load shedding. The annual net energy demand for the system is 119,519 kWh. The contribution from the grid is 95,053 kWh (79.6%). The PV supports 7.2% of the load, supplying 8,688 kWh. The net energy discharged from the BSS is 14,674 kWh (12.3%). From these data, it can be inferred that HES satisfies the total load demand and the proposed EMS strategy works well to maintain the power balance of the system. During the absence of the grid, PV and BSS supply major part of the load while DGen backs the system in emergency situations only. Over the entire year, such situation occurs for 80 hours only. This

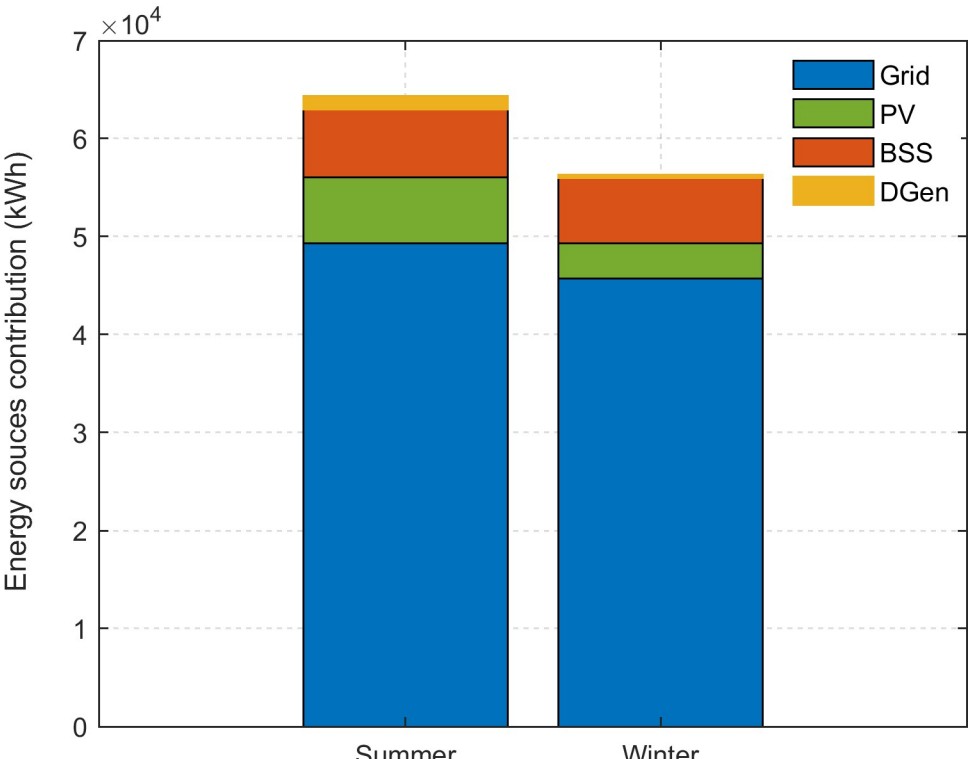

**Fig 16. Contribution of different energy sources in summer and winter season.** Total annual net energy demand: 119,519 kWh.

means that the DGen contributes to less than 1% of overall load. The seasonal contribution from these sources is plotted in Fig 16. Clearly, the contribution of HES is more significant in the summer season due to the higher availability of PV power. During the summer, the day-time is longer: 11 to 12 hours, compared to 7 to 8 hours in winter season [52].

## 3.3 Comparison with the conventional solution

The feasibility of the proposed HES is compared to the existing load shedding mitigation methods namely the UPS and the diesel generator. To be consistent with HES, performance of UPS and generator system is tested for the same load profile (Fig 5). The *LCOE* for these systems is also calculated based on 20 years lifetime like HES. Furthermore, a discount rate of 9% is applied throughout analysis [51]. The generator has the same specification as the one used for HES; also, the diesel price (69 cents/litre) is the same for both cases. For UPS, the total battery capacity is 37.8 kWh with $N_{bat}$ = 21 while for the UPS inverter, similar rating ($P_{inv}$ = 30 kW) as considered for HES is used. Notice that $N_{bat}$ for the UPS has increased compared to optimal design of HES ($N_{bat}$ = 16). This is to fulfil the load requirement solely by the UPS.

The LCOE of the three systems (based on annual simulation results) is shown in Table 4. Among these three options, the HES provides the cheapest electricity. The LCOE for the HES

**Table 4. The LCOE of different energy systems.**

| HES | UPS | Diesel Generator |
|---|---|---|
| (US cents/kWh) | (US cents/kWh) | (US cents/kWh) |
| 8.32 | 13.06 | 29.19 |

is 36.3% and 71.5% cheaper than UPS and generator system, respectively. The results clearly prove the superiority of HES over the conventional methods. Besides for HES, the generator operating hours have significantly reduced (80 hours) compared to only generator system case where generator operation is required during each load shedding interval. Furthermore, the utilization of PV significantly reduces the grid burden by 47.9% and 13.1% compared to the solution using the UPS and DGen, respectively. The PV contribution to support grid operation is explain in the proceeding section.

## 3.4 Performance with feed-in-tariff

It should be emphasized that economic projections provided in Table 4 do not include the FiT incentives. The FiT offers monetary gains for renewable power producers, which strongly influences the operational cost and the economic viability of HES [53]. For Pakistan, the FiT price for one unit of electricity from renewables is 12 US cents. To observe the impact of FiT, the HES operation that depicts the amount of PV energy injected to the grid is shown in Figs 17 and 18. For convenience, the typical summer and winter day scenarios are used for illustration. Notice that application of FiT takes place when the extra energy is injected to the grid via *PV_feed_Grid* operation (yellow area). This occurs when PV is having surplus power during *Grid mode* operation. The HES starts to earn profit due to the difference between the *LCOE* (8.32 cents) and the *FiT* (12 cents) during this interval. The amount of earnings can be quantified as

$$Earnings\,(t) = \sum_{n=1}^{N} \sum_{t=1}^{8784} \left( FiT - LCOE \right) \times PV\_feed\_Grid\,(t) \qquad (35)$$

Based on the received profit, the *LCOE* is re-evaluated and found to be 7.02 US cents/kWh. This is 15.62% lower than the LCOE without FiT. This is an interesting finding and the price is attractive for both customers and investors consideration. With the declining costs of modules and batteries, the HES will be more attractive in future. As the economics of scale and consumer's interest increase, the prices of components will further decline. The decrease in system costs, coupled with the projected increase of grid electricity price, the HES can become a viable solution for the soft power shortage problem in Pakistan.

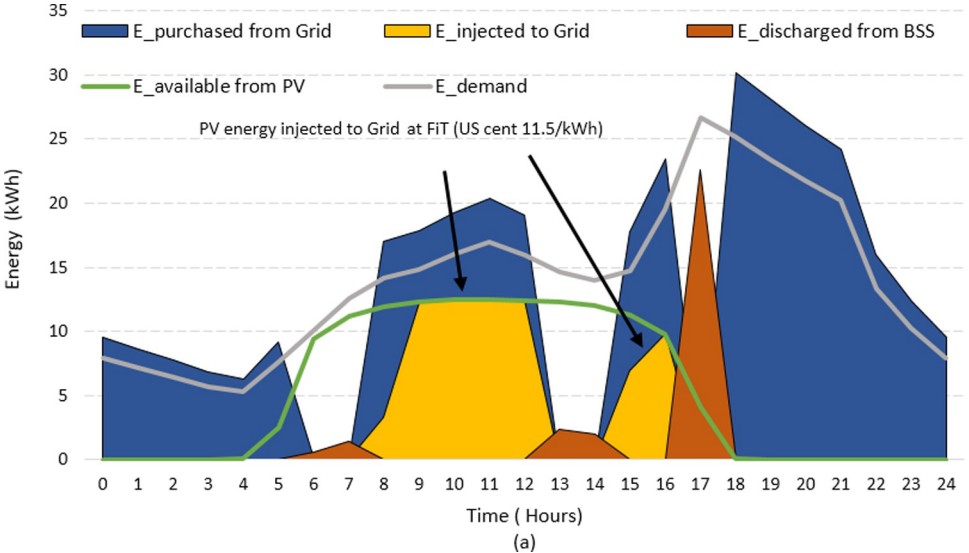

**Fig 17. The impact of FiT during summer day.**

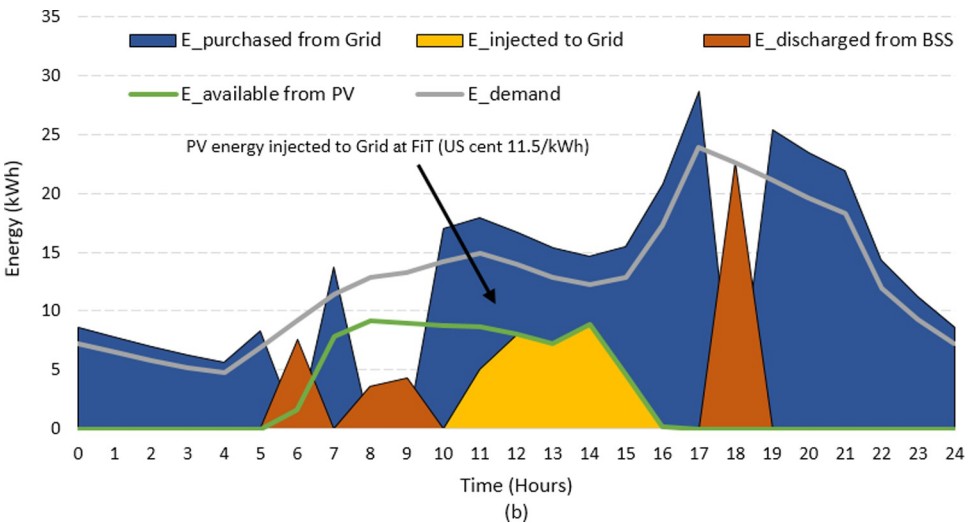

**Fig 18. The impact of FiT during winter day.**

## 4. Conclusion

As the renewables are expected to be the main contributors of electricity generation, research on techno-economic aspects is of practical importance. Due to the existing energy infrastructure in Pakistan, the most reasonable use of renewable energy technologies is small-scale installation to serve load during peak hours and grid-constrained conditions. This paper demonstrated techno-economic modelling of HES to deal with critical problem of load shedding. The modelling approach integrated both the technical and economic aspects of HES to design a reliable and cost effective system. PSO was implemented to determine the optimum size of HES components (PV, BSS) based on the minimum value of the LCOE. One of the strengths of this work is to consider the real load shedding schedule and actual meteorological data of the studied location. Annual simulations were performed for hourly measured data and the results affirmed the reliability of HES through long-term operation with changing operating scenarios. Results showed that LCOE for HES (8.32 cents/kWh) with optimum size is 36.3% and 71.5% cheaper than LCOE of UPS and generator system, respectively (two conventional methods to overcome load shedding in the region). Besides, the incorporation of HES contributed significantly to reduce the burden of the grid. The annual reductions were 47.9% and 13.1% compared to the UPS and generator, respectively. The main conclusion from this work is that Pakistan can utilize renewable based HES to alleviate the existing energy crisis and participate in achieving the Sustainable Development Goal (SDG) 7 that calls for "reliable, affordable, sustainable and modern energy for all". Some guidelines for government and policymakers to facilitate HES implementation can be:

1. Creating an enabling environment and clear policy to encourage private investment in the renewable energy sector.

2. Developing localized institutions to train workers in the skills such as installing, operating and maintaining solar PVs.

3. Accelerating efforts to foster joint research on renewable and conventional systems at the regional level.

## Supporting information

**S1 Data.** https://solcast.com/historical-and-tmy/ **(Meteorological data).**
(XLSX)

**S2 Data.** http://ccms.pitc.com.pk/FeederDetails **(Local feeder load shedding schedule).**
(XLSX)

## Acknowledgments

The author(s) received no specific funding for this work.

## Author Contributions

**Conceptualization:** Muhammad Paend Bakht, Zainal Salam.

**Data curation:** Muhammad Paend Bakht.

**Formal analysis:** Muhammad Paend Bakht.

**Investigation:** Muhammad Paend Bakht.

**Methodology:** Muhammad Paend Bakht, Zainal Salam.

**Project administration:** Zainal Salam.

**Resources:** Zainal Salam, Usman Ullah Sheikh.

**Software:** Usman Ullah Sheikh, Nuzhat Khan, Waqas Anjum.

**Supervision:** Zainal Salam, Abdul Rauf Bhatti.

**Validation:** Nuzhat Khan.

**Writing – original draft:** Muhammad Paend Bakht, Zainal Salam.

**Writing – review & editing:** Zainal Salam, Abdul Rauf Bhatti, Usman Ullah Sheikh.

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
