## [Decision Letter · Decision Letter 0]

4 Jun 2021

PONE-D-21-01708

Techno-economic modeling of hybrid energy system to overcome the load shedding problem: A case study of Pakistan

PLOS ONE

Dear Dr. Bakht,

Thank you for submitting your manuscript to PLOS ONE. After careful consideration, we feel that it has merit but does not fully meet PLOS ONE’s publication criteria as it currently stands. Therefore, we invite you to submit a revised version of the manuscript that addresses the points raised during the review process.

Manuscript need major revision.

We look forward to receiving your revised manuscript.

Kind regards,

Faisal Abbas, PhD

Academic Editor

PLOS ONE

Journal Requirements:

This work is funded by the Higher Education Commission, Pakistan under HRDi-UESTP Scholarship Program and Ministry of Higher Education, Malaysia under the Malaysia Rising Star Award grant. The grant is managed by Universiti Teknologi Malaysia under Vot. No.R.J130000.7823.4F919.

5. Please remove your figures from within your manuscript file, leaving only the individual TIFF/EPS image files, uploaded separately.  These will be automatically included in the reviewers’ PDF.

Additional Editor Comments:

Manuscript need major revisions.

Reviewers' comments:

Reviewer's Responses to Questions

**Comments to the Author**

1. Is the manuscript technically sound, and do the data support the conclusions?

Reviewer #1: Yes

Reviewer #2: No

2. Has the statistical analysis been performed appropriately and rigorously? 

Reviewer #1: Yes

Reviewer #2: No

3. Have the authors made all data underlying the findings in their manuscript fully available?

Reviewer #1: Yes

Reviewer #2: Yes

4. Is the manuscript presented in an intelligible fashion and written in standard English?

Reviewer #1: Yes

Reviewer #2: Yes

5. Review Comments to the Author

Reviewer #1: ABSTRACT: Results of the study need to properly illustrated in the abstract. Currently the abstract mainly highlights the materials and methods used during the study. Please also add the final conclusion and recommendations if any in the abstract

INTRODUCTION: Please properly explain the techno-economic model being used during the study. Objectives of the study are not clear in the introduction. Also please highlight why this research is important and significant.

CONCLUSIONS: Conclusion should properly highlight the key findings of the study and further recommendations

Reviewer #2: The paper presented a techno-economic modelling of hybrid energy system (HES) to maintain an uninterrupted supply of electricity during load shedding for a small residential community situated in the city of Quetta, Pakistan.

The proposed model determines the optimum size of renewable sources based on a minimum levelized cost of electricity. The paper is a technical exercise and a case study that employed the old methodology of LCOE. I have observations regarding the nature of the problem of load shedding in recent times that has changed the debate entirely towards the financial reasons of short supply instead of capacity constraints (that was a major reason of blackouts/ brownouts during last decade). Now that the capacity constraints has been removed by adding plenty of new generation capacity hence, the discussion cannot be completed without taken into account the financial aspect of electricity supply generally termed as the circular debt. Therefore, I think the paper needs major revision by incorporating the financial side.

6. PLOS authors have the option to publish the peer review history of their article (what does this mean?). If published, this will include your full peer review and any attached files.

Reviewer #1: No

Reviewer #2: No

---

## [Author Response · Author response to Decision Letter 0]

29 Sep 2021

Response Letter to Editor

Dear Editor,

We are thankful to you for providing us the opportunity to revise the manuscript (PONE-D-21-01708R1) titled “Techno-economic modelling of hybrid energy system to overcome the load shedding problem: A case study of Pakistan”, after addressing the comments and suggestions by the reviewers. We are also thankful to the respected reviewers for their time and suggesting changes to further improve the quality of the paper. In response, we provide the point to point responses to the comments, presented in the next pages. Kindly note that we have made following changes in the manuscript to fulfill Journal requirements. 

1. Changes have been made in manuscript to follow template and PLOS ONE's style requirements. Figures have been removed from main manuscript copy and submitted separately as Tiff files. Also, all figures have been checked through diagnostic tool ‘PACE’ as directed. Reference style has been amended to meet the requirements.

2. The manuscript has been thoroughly copyedited for language usage, spelling, and grammar by a co-author, Professor Dr Zainal Salam from University Technology Malaysia. Further, proofreading has been done by main author, Muhammad Paend Bakht and a co-author, Nuzhat Khan. 

3. The URL’s of the data applied from public repository have been included as supporting information in the revised manuscript.

4. The funding statement has been amended and aligned with the statement of the online submission form i.e. ‘The author(s) received no specific funding for this work’.

5. Authors contribution has been added in the manuscript. 

We are looking forward for a positive outcome on our manuscript. 

Thank you.

Muhammad Paend Bakht,

Corresponding author.

On behalf of all Authors of Manuscript (PONE-D-21-01708R1) 

Response Letter to Reviewer # 1 

Reviewer #1: 

We are thankful for the comprehensive review and the valuable suggestions to improve the quality of our work. The response to the respected reviewer is given as follows.

Comments from Reviewer #1:

Abstract: Results of the study need to properly illustrated in the abstract. Currently the abstract mainly highlights the materials and methods used during the study. Please also add the final conclusion and recommendations if any in the abstract. 

Introduction: Please properly explain the techno-economic model being used during the study. Objectives of the study are not clear in the introduction. Also please highlight why this research is important and significant.

Conclusions: Conclusion should properly highlight the key findings of the study and further recommendations further recommendations

Response:

As per recommendations by the Reviewer, we have rewritten the Abstract, Introduction and Conclusion section of the paper. In the Abstract part, the results, conclusion and future recommendations are added. The introduction section is revised to highlight research significance. The emphasis has been put to highlight the weakness of existing studies in the current scenario of load shedding. Based on that, the objectives of our study are clearly described in the introduction section. Similarly in the conclusion, key findings of the study and recommendations are included as suggested by the Reviewer. 

Response Letter to Reviewer # 2

Comments from Reviewer #2:

The paper presented a techno-economic modelling of hybrid energy system (HES) to maintain an uninterrupted supply of electricity during load shedding for a small residential community situated in the city of Quetta, Pakistan. The proposed model determines the optimum size of renewable sources based on a minimum levelized cost of electricity. The paper is a technical exercise and a case study that employed the old methodology of LCOE. I have observations regarding the nature of the problem of load shedding in recent times that has changed the debate entirely towards the financial reasons of short supply instead of capacity constraints (that was a major reason of blackouts/ brownouts during last decade). Now that the capacity constraints has been removed by adding plenty of new generation capacity hence, the discussion cannot be completed without taken into account the financial aspect of electricity supply generally termed as the circular debt. Therefore, I think the paper needs major revision by incorporating the financial side.

Response:

We are thankful for the positive comments and valuable suggestions to improve the quality of our work. Please find the response as follows.

The observation with regard to the ‘circular debt’ is highly acknowledged. In response, we revised entire Introduction of paper to include this aspect. In that section, we discussed the roots of Pakistan energy crisis, starting from the 1990’s. In technical terms, the crisis can be divided into two scenarios: the ‘hard power shortage’ and ‘soft power shortage’. The hard power shortage refers to insufficient capacity, in which the required demand for electricity cannot be fulfilled even with the full utilization of the available power generators. On the other hand, soft power shortage is not due to the insufficient installed capacity; rather, it is a result of under-utilization of the power generating units due to the lack of supporting facilities. 

There are various reasons for both shortages. In the final analysis, we concur with the Reviewer that for Pakistan, the recent energy crisis scenario is more inclined towards the soft power shortage. The main reason is because the government is focused on increasing the generation capacity, but very little attention is given to upgrade entire electricity infrastructure, namely the transmission and distribution. Furthermore, there is lack of effort to improve energy mix and system efficiency. Finally, as correctly pointed out by the Reviewer, there is a need to relook into the costs recovery mechanism to minimize the circular debt. All these factors, adds to Pakistan problem of energy sufficiency and security.

Cleary, considering the above-mentioned issues, the short-term electricity shortage remains an resolved problem for the end users. As a result, multiple hours of scheduled load shedding is still a common practice by utility operators. Since the permanent solution is not expected in near future, the scope of this work is to deal with the present load shedding problem using renewables, i.e., the proposed HES. This is localized deployment (at distribution feeder level) and can be considered as a medium term solution. By incorporating the HES in to the system, the users are able to experience continuous electricity supply with reasonable price. It is envisaged that, the widespread use of HES will give the government some breathing space to work on the long term solution, i.e. to expand upgrade existing infrastructure and financing facilities. 

Thus, the HES implementation allows manageability and complementarity of energy that, when applied to existing infrastructure can eliminate or diminish the problem of load shedding. Furthermore, HES deployment in the current scenario of Pakistan is also aligned with governments plan to achieve 30% power generation from renewable energy. This explanation has been thoroughly provided in the introduction section of paper. 

We try our best to address the valuable concerns. All the authors are grateful to the reviewers for sparing the valuable time to upgrade the quality of our manuscript.

---

## [Decision Letter · Decision Letter 1]

9 Feb 2022

PONE-D-21-01708R1Techno-economic modelling of hybrid energy system to overcome the load shedding problem: A case study of PakistanPLOS ONE

Dear Dr. Bakht,

Thank you for submitting your manuscript to PLOS ONE. After careful consideration, we feel that it has merit but does not fully meet PLOS ONE’s publication criteria as it currently stands. Therefore, we invite you to submit a revised version of the manuscript that addresses the points raised during the review process.

Minor Revision Please submit your revised manuscript by Mar 26 2022 11:59PM. If you will need more time than this to complete your revisions, please reply to this message or contact the journal office at plosone@plos.org. Please include the following items when submitting your revised manuscript:A rebuttal letter that responds to each point raised by the academic editor and reviewer(s). You should upload this letter as a separate file labeled 'Response to Reviewers'.A marked-up copy of your manuscript that highlights changes made to the original version. You should upload this as a separate file labeled 'Revised Manuscript with Track Changes'.An unmarked version of your revised paper without tracked changes. You should upload this as a separate file labeled 'Manuscript'.If applicable, we recommend that you deposit your laboratory protocols in protocols.io to enhance the reproducibility of your results. Protocols.io assigns your protocol its own identifier (DOI) so that it can be cited independently in the future. For instructions see: https://journals.plos.org/plosone/s/submission-guidelines#loc-laboratory-protocols. Additionally, PLOS ONE offers an option for publishing peer-reviewed Lab Protocol articles, which describe protocols hosted on protocols.io. Read more information on sharing protocols at https://plos.org/protocols?utm_medium=editorial-email&utm_source=authorletters&utm_campaign=protocols.

We look forward to receiving your revised manuscript.

Kind regards,

Faisal Abbas, PhD

Academic Editor

PLOS ONE

Journal Requirements:

Additional Editor Comments (if provided):

Minor Comments.

Reviewers' comments:

Reviewer's Responses to Questions

**Comments to the Author**

1. If the authors have adequately addressed your comments raised in a previous round of review and you feel that this manuscript is now acceptable for publication, you may indicate that here to bypass the “Comments to the Author” section, enter your conflict of interest statement in the “Confidential to Editor” section, and submit your "Accept" recommendation.

Reviewer #1: All comments have been addressed

Reviewer #3: All comments have been addressed

2. Is the manuscript technically sound, and do the data support the conclusions?

Reviewer #1: Yes

Reviewer #3: Yes

3. Has the statistical analysis been performed appropriately and rigorously? 

Reviewer #1: Yes

Reviewer #3: Yes

4. Have the authors made all data underlying the findings in their manuscript fully available?

Reviewer #1: Yes

Reviewer #3: Yes

5. Is the manuscript presented in an intelligible fashion and written in standard English?

Reviewer #1: No

Reviewer #3: Yes

6. Review Comments to the Author

Reviewer #1: Author addressed most of my comments, however, it still requires improvements.

1. Abstract should include more results in accordance with the results and discussions section.

2. Overall structure of the manuscript should be changed including clearly defined headings Introduction, Material and Methods, Results and discussions. Heading 2 and onward should be the part of introduction under heading 1.

3. Initial lines (Line 437-450) of results and discussion section should be the part of materials and methods section, which is currently missing in the manuscript.

4. Results need to be properly supported by the relevant literature. Currently very small amount of literature support is seen in results and discussion section.

4. Conclusion section still missing key findings of the study. Please provide some important results of the study in conclusion section.

5. Overall quality of English (grammar, spellings, sentence structure etc.) needs significant improvements

Reviewer #3: The paper focuses on a very interesting and urgent topic.

The motivation of this study is weak and author can mention the motivation of the study in the introduction section.

How this research can be used in practice (economic and societal impact) and to influence public policy (contributing to the body of knowledge)? This can be written in the introduction.

In the conclusion section, the author should identify clear implications for research, practice and/or society. Currently, this part is weak.

7. PLOS authors have the option to publish the peer review history of their article (what does this mean?). If published, this will include your full peer review and any attached files.

Reviewer #1: No

Reviewer #3: No

---

## [Author Response · Author response to Decision Letter 1]

24 Mar 2022

Response to Reviewers 

Reviewer #1: 

We are thankful for the comprehensive review and the valuable suggestions to improve the quality of our work. The response to the respected reviewer is highlighted in yellow color in the manuscript. For convenience, the point-to-point response and its exact location in the manuscript is given as follows.

Comment 1: Abstract should include more results in accordance with the results and discussions section. The entire abstract is revised and main results have been included. Lines 14-28 

Overall structure of the manuscript should be changed including clearly defined headings Introduction, Material and Methods, Results and discussions. Heading 2 and onward should be the part of introduction under heading 1. Response 1: The entire manuscript structure is changed as suggested. Kindly note that extra text and Figure 1 are added to clear the overall methodology of the paper in Section 2. (Lines 121-134)

Comment 2: Initial lines (Line 437-450) of results and discussion section should be the part of materials and methods section, which is currently missing in the manuscript. The lines have been shifted to materials and methods section as recommended. Lines 338-342

Results need to be properly supported by the relevant literature. Currently very small amount of literature support is seen in results and discussion section. Response 2: To support the results, relevant studies have been included in the results section. Cited studies are Reference [49] [20,50,51][53]. Extra text is added to justify the results in lines (500-503). 

(Lines 493-501)

Comment 3: Conclusion section still missing key findings of the study. Please provide some important results of the study in conclusion section. Response 3: The entire conclusion is revised and the relevant results have been included. Some implications and recommendations are also added. 

 (Lines 603-607)

Comment 4: Overall quality of English (grammar, spellings, sentence structure etc.) needs significant improvements Response 4: We have proof read and tried to address small errors in the manuscript. Alos, Equations and Figure numberings have been corrected. (Multiple lines)

Reviewer #2: 

We are thankful for the comprehensive review and the valuable suggestions to improve the quality of our work. The response to the respected reviewer is highlighted in green color in the manuscript. For convenience, the point-to-point response and its exact location in the manuscript is given as follows.

Comment 1: The paper focuses on a very interesting and urgent topic. The motivation of this study is weak and author can mention the motivation of the study in the introduction section. Response 1: First, we thank you for the positive comment, which has encouraged us. As suggested, we have revised the introduction section. A new paragraph in the introduction section is added to identify the problem and the research significance to address the problem. Other important justifications have also been included. (Lines 32-44, 107-108, 115-116)

Comment 2: How this research can be used in practice (economic and societal impact) and to influence public policy (contributing to the body of knowledge)? This can be written in the introduction. Response 2: The economic and societal impacts of the research are highlighted. The impact of the research on public policy is highlighted as recommended. Lines (75-79, 103-105, 107-108, 115-116)

Comment 3: In the conclusion section, the author should identify clear implications for research, practice and/or society. Currently, this part is weak. Response 3: The conclusion section is revised to identify the implications of the carried research. Besides, Lines (592-595, 607-617)

We try our best to address the valuable concerns. All the authors are grateful to the reviewers for sparing the valuable time to upgrade the quality of our manuscript.

---

## [Editor Report · Decision Letter 2]

25 Mar 2022

Techno-economic modeling of hybrid energy system to overcome the load shedding problem: A case study of Pakistan

PONE-D-21-01708R2

Dear Dr. Bakht,

We’re pleased to inform you that your manuscript has been judged scientifically suitable for publication and will be formally accepted for publication once it meets all outstanding technical requirements.

Kind regards,

Faisal Abbas, PhD

Academic Editor

PLOS ONE

Additional Editor Comments (optional):

Accept
---

## [Editor Report · Acceptance letter]

14 Apr 2022

PONE-D-21-01708R2 

Techno-economic modelling of hybrid energy system to overcome the load shedding problem: A case study of Pakistan 

Dear Dr. Bakht:

I'm pleased to inform you that your manuscript has been deemed suitable for publication in PLOS ONE. Congratulations! Your manuscript is now with our production department. 

Kind regards, 

on behalf of

Dr. Faisal Abbas 

Academic Editor

PLOS ONE